Manuscript prepared for Earth Syst. Dynam.
with version 2015/04/24 7.83 Copernicus papers of the LATEX class copernicus.cls.
Date: 21 June 2016

# Comment on "Scaling regimes and linear/nonlinear responses of last millennium climate to volcanic and solar forcing" by S. Lovejoy and C. Varotsos

Kristoffer Rypdal[1] and Martin Rypdal[1]

[1]Department of Mathematics and Statistics, UiT The Arctic University of Norway, Norway
*Correspondence to:* Kristoffer Rypdal  (kristoffer.rypdal@uit.no)

**Abstract.** *Lovejoy and Varotsos* (2016) (L&V) analyse the temperature response to solar, volcanic, and solar plus volcanic, forcing in the Zebiak-Cane (ZC) model, and to solar and solar plus volcanic forcing in the GISS-E2-R model. By a simple wavelet filtering technique they conclude that the responses in the ZC model combine subadditively on time scales from 50 to 1000 yr. Nonlinear response on shorter time scales is claimed by analysis of intermittencies in the forcing and the temperature signal for both models. The analysis of additivity in the ZC model suffers from a confusing presentation of results based on an invalid approximation, and from ignoring the effect of internal variability. We present tests without this approximation which are not able to detect nonlinearity in the response, even without accounting for internal variability. We also demonstrate that internal variability will appear as subadditivity if it is not accounted for. L&V's analysis of intermittencies is based on a mathematical result stating that the intermittencies of forcing and response is the same if the response is linear. We argue that there are at least three different factors that may invalidate the application of this result for these data. It is valid only for a power-law response function, it assumes power-law scaling of structure functions of forcing as well as temperature signal, and the internal variability, which is strong at least on the short time scales, will exert an influence on temperature intermittence which is independent of the forcing. We demonstrate by a synthetic example that the differences in intermittencies observed by L&V easily can be accounted for by these effects under the assumption of a linear response. Our conclusion is that the analysis performed by L&V does not present valid evidence for a detectable nonlinear response in the global temperature in these climate models.

## 1 Introduction

The issue of linearity in the global temperature responses of modern General Circulation Models (GCMs) and Earth System Models (ESMs) is important because the prospect of predicting global aspects of the climate under different forcing scenarios is considerably brighter if the response is reasonably linear. Linear response models with two characteristic response times or a long-memory

power-law response have had considerable success in describing global temperature response in GCM data, instrumental data and in multiproxy reconstructions (*Held et al.*, 2010; *MacMynowski et al.*, 2011; *Geoffroy et al.*, 2013; *Caldeira and Myhrvold*, 2013; *Rypdal and Rypdal*, 2014; *Østvand et al.*, 2014; *Rypdal et al.*, 2015; *Lovejoy et al.*, 2015; *Fredriksen and Rypdal*, 2016). The credibility of these results depends crucially on the validity of the linear approximation in the global response. Particularly relevant is *Geoffroy et al.* (2013), who estimate the parameters of a linear two-box energy balance model by data from runs of a large number of CMIP5 ESMs with step-function forcing and linearly increasing forcing, respectively. Very good fits to the simulated global temperature are found in this study, with the same values of the two-box model parameters for the two different forcing scenarios. This is a very clear demonstration of the approximate linearity of the global temperature response in the CMIP5 ensemble. The issue of additivity of the temperature response in GCMs has been extensively studied over the last two decades, and the majority of studies find only weak nonlinearities in the global response, although nonlinearites are often found in regional responses in some models (*Ramaswamy and Chen*, 1997; *Meehl et al.*, 2004; *Kirkevåg*, 2008; *Shiogama et al.*, 2013).

The paper by *Lovejoy and Varotsos* (2016) (in the following denoted L&V) is a research paper, but has the character of a review of earlier papers of Shaun Lovejoy and coworkers. The review style has the unfortunate effect of masking the substance of the new results presented, which is an analysis of the responses in two different climate models to solar and volcanic forcing, and to combinations of these forcings. The actual analysis is made in Section 3.4 of the L&V paper, where the authors test the additivity of responses to solar and volcanic forcing in the Zebiak-Cane (ZC) model, and in Section 4.2, where they study the intermittency of forcing and response and conclude that difference in their intermittency implies nonlinearity of the response. In Sect. 2 of this comment we present a critical examination of the methods L&V invoke to conclude that combined solar and volcanic forcing leads to a weaker response than the sum of the solar and volcanic responses in the ZC model. Sect. 3 examines the intermittency analysis and demonstrates that L&V's results for the ZC model can be reproduced in the response of a simple linear response model. In Sect. 4 we present some personal views in the light of the interactive discussion on this comment, and Sect. 5 contains our main conclusions.

## 2  Linearity and response additivity

### 2.1  The logic of hypothesis testing

According to a widely accepted principle in the philosophy of science (*Popper*, 1959), a well-posed scientific hypothesis has to be *falsifiable* by experiment or observation. There is an infinity of ways the temperature response can be nonlinear. This pertains to both details of the nonlinear interactions and to their magnitude. No test is infinitely accurate, so there will always be a possibility that a weak

nonlinearity goes undetected. Hence, is not logically possible to formulate a falsifiable hypothesis stating that the response is nonlinear. The well-posed hypothesis is that the response is linear. From this hypothesis one can design tests by which the hypothesis can be rejected by conceivable outcomes of experiments or observations. If such a test fails to reject the linearity hypothesis, we cannot conclude that the response is linear, but if a series of increasingly sharper tests still fail to reject the linearity hypothesis will stand stronger. This is the principle of induction. On the other hand, if a test turns out to reject linearity, then we have detected a nonlinearity. So, even though nonlinearity cannot be falsified, it can in fact be verified. This is because nonlinearity is the negation of the falsifiable linearity hypothesis; if a statement $A$ is false then the statement *not A* is true.

Based on this logic, the only reasonable approach is to formulate a test that may, or may not, reject the hypothesis that the response is linear. The hypothesis, however, must be formulated with some care. The issue in the L&V paper is nonlinearity in the response of hydrodynamic flow models like the ZC and GCMs, which are known to be inherently nonlinear. It is not difficult to devise tests that will detect nonlinearities in these models. The question at hand, however, is not whether nonlinearities are present, but whether these nonlinearities are detectable in the global temperature response.

In GCM-type models "unforced" control simulations are of course driven by the constant solar energy flux, and this results in a turbulent, nonlinear cascade that forms the "internal variability" of the model. In a linear model for the global response this internal variability is represented as a noise process $\epsilon(t)$ in a global variable $T(t)$. Forcing $F(t)$ in the model means a variation of the global energy flux around the flux that drives such a turbulent equilibrium state.

## 2.2 The linear response hypothesis

After these remarks we are ready to formulate the *linear response hypothesis*:

(i) For realistic strength of the global forcing the the statistics of the internal variability $\epsilon(t)$ is unaffected by the forcing.

(ii) The global temperature can be expressed as a sum of this internal variability and a linear response to the forcing, i.e.,

$$T(t) = T^{\text{det}}(t) + \epsilon(t), \;\; T^{\text{det}}(t) = \hat{L}[F(t)], \tag{1}$$

where $T(t)$ is the global surface temperature, $T^{\text{det}}(t)$ is the deterministic, linear response to the global forcing $F(t)$, and $\hat{L}$ is the linear response operator.

## 2.3 Internal noise and response additivity

The data used from the ZC model is the temperature (more precisely; the Niño3 index) after averaging over 100 simulations with the same forcing. If the internal variability is a persistent noise, averaging over $N$ independent runs will reduce the standard deviation by a factor $N^{-1/2} = 0.1$, but

the correlation structure of the noise will be preserved. In the following, $\epsilon(t)$ is the noise that remains
    after averaging the internal noise over those $N$ realisations.

    The next step is to produce a fluctuation $\Delta T(t, \Delta t)$ by means of a linear low-pass filtering op-
    eration. It could for example be a simple moving average over a window $\Delta t$, or the Haar wavelet
    smoothing employed by L&V. In the following we shall for notational simplicity omit the arguments
$(t, \Delta t)$. The results presented hold for the temperature signal itself ($\Delta t = 0$) as well as for any degree
    $\Delta t$ of filtering. Since the response operator $\hat{L}$ is linear we have

$$\Delta T_{v+s}^{\text{det}} = \Delta T_{s}^{\text{det}} + \Delta T_{v}^{\text{det}}, \tag{2}$$

    where $\Delta T_{s}^{\text{det}}$ and $\Delta T_{v}^{\text{det}}$ are the responses to the solar and volcanic forcings, $\Delta F_s$ and $\Delta F_v$, respec-
    tively, and $\Delta T_{v+s}^{\text{det}}$ is the response to the combined forcing $\Delta F_s + \Delta F_v$. This yields

$$\Delta T_s = \Delta T_{s}^{\text{det}} + \Delta \epsilon_s, \tag{3}$$

$$\Delta T_v = \Delta T_{v}^{\text{det}} + \Delta \epsilon_v, \tag{4}$$

$$\Delta T_{s+v} = \Delta T_{s}^{\text{det}} + \Delta T_{v}^{\text{det}} + \Delta \epsilon_{s+v}. \tag{5}$$

    Here $\Delta \epsilon_s$, $\Delta \epsilon_v$, and $\Delta \epsilon_{s+v}$ are the filtered fluctuations of independent realisations of the same noise
    process $\epsilon(t)$ (here $\epsilon(t)$ is the average over 100 realisations of internal variability). By subtracting
Eqs. (3) and (4) from Eq. (5), and using Eq. (2), we find

$$\Delta T_{s+v} - \Delta T_s - \Delta T_v = \Delta \epsilon_{v+s} - \Delta \epsilon_s - \Delta \epsilon_v \equiv \Delta \varepsilon. \tag{6}$$

    Here, $\Delta \varepsilon$ is the sum of three independent realisations of the same noise process $\Delta \epsilon \overset{d}{=} \Delta \epsilon_s \overset{d}{=} \Delta \epsilon_v \overset{d}{=}$
    $\Delta \epsilon_{v+s}$, where $\overset{d}{=}$ is identity in distribution. This imples that

$$\Delta \varepsilon \overset{d}{=} \sqrt{3} \Delta \epsilon. \tag{7}$$

Hence, a prediction based on the linear response hypothesis is that the difference between the tem-
    perature driven by combined solar and volcanic forcing and the sum of the temperatures driven by
    solar and volcanic forcing is realisation of a noise process which is $\sqrt{3}$ times the internal variability
    process. In Sect. 2.4 we shall test this prediction on the data from the ZC model. If the prediction is
    inconsistent with the data the linear response hypothesis is rejected for this model, and nonlinearity
in the response has been detected. If the prediction is confirmed by the data the linear hypothesis
    stands stronger.

## 2.4 Alternative test of additivity in the ZC model

    Fig. 1a shows time series of the solar and volcanic forcing for the last millennium used in the simu-
    lations of the ZC model. Unfortunately L&V did not have available control runs on millennial scale
from this model. This would have been very useful in establishing directly the statistical properties
    of the internal noise $\epsilon(t)$. The approach we will use as an alternative, is to assume the validity of

the linear response hypothesis, which will allow us to extract the internal noise from the simulation with solar forcing only. Then we will formulate a test by which the hypothesis could be rejected by the data for volcanic forcing only and for volcanic plus solar forcing. Assuming the validity of the linear hypothesis from the start may seem like circular reasoning, but it is not. Any valid hypothesis testing makes predictions based on the hypothesis, which are then tested against observation.

If the linear response hypothesis is true we can determine $\epsilon(t)$ from the solar forcing signal and the corresponding temperature signal. The solar forcing signal in Fig. 1a has a smooth appearance, in particular for the first 750 years of the record, for which no sunspot counts were available. As a contrast, the corresponding temperature signal shown as the thin orange curve in Fig. 1b is noisy on all scales down to the annual scale. This appearance of the temperature signal under the smooth solar forcing already lends support to the assumption that the variability up to century time scale is internal. However, according to L&V the subadditivity is most prominent on time scales longer than 50 yr, so we have to pay special attention to the slow components of the noise spectrum. We now write a linear response to the solar forcing in the form;

$$\Delta T_s^{\text{det}}(t,\Delta t) = -S\,\Delta F_s(t-\tau,\Delta t). \tag{8}$$

Here $\Delta t = 50$ yr over which we have performed a moving average of the temperature and forcing. The time lag $\tau$ of the response is estimated to be $\approx 25$ yr from inspection of the filtered time series. The climate sensitivity $S$ is chosen to give the best least-square fit of $\Delta T_s^{\text{det}}(t,\Delta t)$ (the black curve in Fig. 1b) to the filtered temperature signal $\Delta T_s(t,\Delta t)$ (the orange, thick curve).

Because of the smooth character of the solar forcing signal in the first 750 yr of the record, the 50 yr filtering of this signal has almost no effect, and we can therefore interpret the black curve in Fig. 1b as the linear, deterministic response to the solar forcing, and the difference between the orange, thin curve and the black curve as the internal noise, i.e.,

$$\epsilon(t) = T_s(t) - \Delta T_s^{\text{det}}(t,\Delta t). \tag{9}$$

This difference is plotted as the brown, thin curve at the bottom of Fig. 1b, and the thick brown curve is the 50 yr moving average.

We have now distinguished the internal noise from the solar-driven temperature signal by means of the very simple linear-response assumption, Eq. (8). This response function is of course not accurate, the delay in the response should rather be expressed as a time-dependent response function (a frequency-dependent transfer function) rather as a fixed delay (*Rypdal and Rypdal*, 2014). For the ZC model we do not have detailed information about the response function, so we have no means of constructing one that is known to be better than Eq. (8). But for the present purpose this is not crucial since the solar forcing has almost no power in the high frequencies.

The orange bullets in Fig. 1d is a characterisation of this noise by means of the Haar structure function employed by L&V. The definition of this structure function is

$$\sqrt{S_2^{\text{Haar}}(\Delta t)} = \langle |\Delta T(t,\Delta t)|^2 \rangle^{1/2}, \tag{10}$$

where $\langle \ldots \rangle$ denotes averaging over disjoint time intervals of length $\Delta t$. It measures the root-mean-square fluctuation level on the scale $\Delta t$. The flat appearance on scales above a decade indicates a

strongly persistent noise process with equally strong fluctuations on scales $\Delta t > 10$ yr. The straight-line character of the log-log plot in this scale range is symptomatic of a scaling process, and the corresponding power spectral density has the form $\sim f^{-\beta}$, where $\beta \approx 1$ (sometimes denoted $1/f$-noise or pink noise). The higher fluctuations for $\Delta t < 10$ yr is characteristic for the El Niño Southern Oscillation (ENSO). This mode is particularly strong in the ZC model, which is designed specifically

for the study of ENSO, and the global output $T(t)$ is the so-called Niño3 index.

If the characterisation we have made of the internal noise is correct, and the linear hypothesis is true, then Eq. (7) must be true. But $\varepsilon$ in Eq. (7) must be computed from Eq. (6), which requires the temperature signals $T_v$ and $T_{u+v}$, in addition to $T_s$. The characterisation of $\epsilon$ only used $T_s$, so if the linear hypothesis is false, it is very unlikely that the estimated $\varepsilon$ and $\epsilon$ will give good agreement with

Eq. (6). This means that we should have a strong test.

In Fig. 1c the thin, blue curve represents $T_{s+v}(t)$, the thin, red curve is $T_s(t) + T_v(t)$, and the thin, black curve is their difference $\varepsilon(t) = T_s(t) + T_v(t) - T_{u+v}$. Note that the narrow spikes from the fast responses to the volcanic eruptions are completely absent in the difference signal $\varepsilon(t)$, demonstrating that the addition of solar forcing does not exert a detectable influence on the response to the volcanic

eruptions on the short time scales up to a few years. The thick curves in Fig. 1c are the corresponding 50 yr moving averages. The Haar structure function of the signal $\varepsilon(t)$ is shown as the red bullets in Fig. 1d. The brown bullets are $\sqrt{3}\epsilon(t)$, i.e., the orange bullets multiplied by $\sqrt{3}$. We observe that the red and brown bullets are more or less on top of each other; the two curves are entangled for $\Delta t > 10$ yr. This means that the second-order statistics of the noise processes $\varepsilon(t)$ and $\sqrt{3}\epsilon$ are

indistinguishable, in agreement with Eq. (7). Thus, this test is not able to reject the linear response hypothesis.

This test would have been stronger if we had a more direct estimate of the internal variability. In an interactive comment, *Lovejoy et al.* (2016) suggest to use a different estimate of the internal noise, namely the first 195 yr of the volcanic-driven response time series. This is justified, since

there was no volcanic forcing in this period. The drawback, however, is that an estimate of the Haar fluctuation from such a short time series is associated with higher estimation uncertainty (finite-sample size errors). Unfortunately, they make no attempt to demonstrate that the estimates of the difference $|\sqrt{3}\epsilon(t) - \varepsilon(t)|$ is significantly different from zero in a statistical sense. Such a test is easy to make by creating a Monte Carlo ensemble of time series containing 195 data points with

statistical properties similar to those of the observed volcano response. The statistical scatter of the Haar fluctuations within this ensemble will give us information about the finite sample uncertainty of the Haar estimate. This is done in Fig. 2, where the specifications of the Monte Carlo ensemble are described in the caption. The figure shows that the difference between the Haar fluctuations of $\sqrt{3}\epsilon(t)$ and $\varepsilon(t)$ is smaller than this uncertainty in the interesting scale range $\Delta t > 10$ yr. This means

that the deviation from linearity observed is statistically insignificant, and hence does not reject the linear response hypothesis. A similar Monte Carlo ensemble for 1000 yr long time series would reduce the scatter in the Haar fluctuations by approximately a factor $??195/1000 ? 0.44$, which is still large enough to conclude that the difference between the blue and brown bullets in Fig.1d is not statistically significant.

## 2.5 Examination of L&V's test of response additivity

The L&V test of additivity shown in their paper is is simpler than described in Sec. 2.4, but ignores internal variability. Here we shall demonstrate that their test also fails to reject the linearity hypothesis, even when this variability is not taken into account. Their main conclusion concerning additivity of responses in the ZC model is that for $\Delta t > 50$ yr the rms-ratio,

$$R \equiv \sqrt{\frac{\langle |\Delta T_s + \Delta T_v|^2 \rangle}{\langle |\Delta T_{u+v}|^2 \rangle}}, \tag{11}$$

is found to be $R \approx 1.5$. As will be shown below, our analysis yields a number indistinguishable from unity. But the authors also make attempts in their Figure 3 to inflate this ratio further by presenting results for the numerator based on the flawed approximation of neglecting the estimate of $\langle \Delta T_s \Delta T_v \rangle$. The approximation is flawed because, even though solar and volcanic forcing are independent processes, the ensemble average $\langle \ldots \rangle$ is estimated from only one realisation of each of these forcing processes. On the short time scales the approximation makes sense, since the ensemble average is replaced by time averages, but as the time scales $\Delta t$ approaches the length of the time series, the number of independent time windows to average over goes to zero. In their Figure 3b L&V show the flawed graph of $\sqrt{\langle |\Delta T_s + \Delta T_v|^2 \rangle}$ based on this approximation together with the graph of $\sqrt{\langle |\Delta T_{u+v}|^2 \rangle}$, which appears to show that the former is larger than the latter by a factor $\approx 2.5$ for $\Delta t > 50$ yr. In Fig. 3 we show the results that we obtain without the approximation.We cannot find any significant difference between the two graphs (red and blue bullets) for $\Delta t < 300$ yr; the two curves are entangled, just as in Fig. 1d. For $\Delta t > 300$ yr the the observed differences are clearly not statistically significant.

An alternative, and very simple, estimate for this ratio can be obtained from the data for the red and blue, thick curves in Fig. 1c, by computing $\Delta T$'s as 50 yr moving averages rather than Haar fluctuations. The standard deviation of $\Delta T_s + \Delta T_v$ is 0.072 K, and of $\Delta T_{s+v}$ it is 0.060 K, which yields $R \approx 1.20$. This ratio is slightly greater than unity due to the higher fluctuations in the red graph compared to the blue graph in Fig. 3 for $\Delta t > 300$ yr. Since this difference on the longest time scales appears to be a statistical error due to limited sample size, $R = 1$ is within the error bars of the estimated $R$ (on these time scales there are only a few independent samples available for estimation of the variance). If such an error test were crucial, we could have computed the uncertainty range via a Monte Carlo ensemble of the $1/f$ noise process, like we did in Fig. 2. However, since the two curves are entangled for $\Delta t > 10$ yr, even very small finite sample size uncertainty will not allow

us to decide that one signal has more power than the other. Moreover, as will be shown in Sect. 2.6, internal variability gives an additional positive contribution to $R$ which exceeds the error that is required to explain the estimate $R \approx 1.2$ under the linear response hypothesis.

### 2.6 The effect of internal variability on the L&V test

The ratio $R$ defined in Eq. (11) only measures the ratio of responses if the internal noise is negligible. Hence, even if $R$ were significantly (in statistical sense) greater than unity, this increase might be caused by the internal variability in a model whose response to forcing is perfectly linear. By using Eq. (6), which is valid for a linear response model, Eq. (11) can be written as

$$R = \sqrt{1 + \frac{\langle|\Delta\varepsilon|^2\rangle}{\langle|\Delta T_{s+v}|^2\rangle}}. \tag{12}$$

This shows that internal noise can increase the rms-ratio computed by L&V even if the response is linear. From the data for the thick, brown curve in Fig. 1b we have that the standard deviation for the internal noise $\Delta\epsilon$ is 0.03, and hence for $\Delta\varepsilon$ a factor $\sqrt{3}$ larger. The standard deviation of $\Delta T_{s+v}$ can be estimated from the data for the thick, blue curve in Fig. 1c and is 0.06. This yields $\langle|\Delta\varepsilon|^2\rangle/\langle|\Delta T_{s+v}|^2\rangle \approx 0.75$, and hence $R \approx 1.32$ is the estimate of the rms-ratio based on the linear response hypothesis.

### 2.7 L&V's arguments against high internal variability

In the first and second drafts of the L&V discussion paper internal variability was not mentioned. After this problem was raised by us in the interactive discussion, L&V have in the final paper presented two arguments against the presence of sufficiently high internal fluctuations on the centennial time scales to explain the raised rms-ratio $R$.

The first argument uses the internal variability of the GISS model as an estimate of the centennial scale internal variability of the ZC model, and concludes that this estimate is less than 20% of the total variability in the ZC model. The authors overlook the fact that the output of the ZC model is the Niño3 index (temperature anomalies in the tropical pacific), while the GISS model output is the average over the northern hemisphere land. One should also keep in mind that the ZC model was never intended to get the statistics of variability correct, and so there is no basis for assuming anything about the magnitude of it relative to GISS. In Figure 4 of the L&V paper, fluctuation levels versus scale for ZC and GISS are plotted in the same panel. For $\Delta t > 10$ yr they almost overlap. However, the ZC model data are averaged over 100 model runs, so the actual fluctuation level for the stochastic component is ten times greater than for the output from GISS control simulations.

The second argument assumes that the internal noise must have a scaling exponent $\beta \approx 0.6$, which would yield a negative slope $H = (\beta - 1)/2 \approx -0.2$ of the structure-function plot (see Fig. 1d). The actual plot of the structure function of the solar residual (the yellow circles in Fig. 1d) has a weakly positive slope, and hence the authors conclude that the latter is dominated by forced fluctuations on

the centennial to millennium scale. The weakness of this argument is that it takes as assumption what the authors want to prove, namely that internal fluctuations on long time scales are small. It seems that only long control runs of the ZC model can settle this issue.

## 2.8 Additivity in NorESM data

There are at least three drawbacks with the ZC data. The model is not representative for the global temperature response, the data analysed has been averaged over 100 realisations, and L&V had no control runs available to assess the magnitude of internal variability. They also analysed data from the NASA GISS-E2-R model, but here they lacked the full suite of simulations with solar-only, volcanic only, and solar+volcanic forcing, and hence they could not perform the test of the additivity of responses on a full-blown GCM. We have acquired a full suite of millennium-long simulations for the NorESM Earth System Model, which is part of the CMIP5 ensemble. More specifically, we have analysed solar-only, volcanic only, solar+volcanic+anthropogenic, and control runs for the 900 yr period 935-1834 CE. We have omitted the period after 1835 CE to minimize the anthropogenic forcing in the full forcing simulation, and treat this as a solar+volcanic simulation. It is remarkable that all Haar fluctuation curves of all these signals are almost flat, corresponding to $H \approx 0$ or $\beta \approx 1$, i.e., to a so called $1/f$-noise.

In Fig. 4 we have plotted the Haar fluctuations for the solar+volcanic (total) forcing (red), for the summed responses to solar and volcanic forcing (blue), and for the control run (magenta). Observe that the responses to solar and volcanic forcing add up to the response of the combined forcing. The subadditivity claimed by L&V is completely absent. We also observe that the internal variability represented by the control run is quite strong. The standard deviation of the internal variability is 2/3 of the variability of the signal with solar+volcanic forcing. Moreover, the internal fluctuations are almost equally strong on long-time scales as on short time scales, contrary to what has been claimed by L&V.

## 3 Linearity and intermittencies

The essence of Section 4 in the L&V paper is a mathematical result claiming that linearity in the response implies that the intermittency (the curvature of the scaling function) is the same for forcing and response. We have a number of reservations against the application of this result to the data and the climate models studied in this paper.

### 3.1 The essence of our critique

There are at least three possible sources of different intermittencies of the forcing and temperatures that are missed in the L&V paper:

(I) The mentioned mathematical result depends on a power-law form of the linear response function. On time scales less than a few years, GCM responses appear to be exponential rather than power-law, as shown for the GISS-ER-2 model in in Fig. 5. On the long time scales this assumption is in direct contradiction to L&V's own claim that GCMs do not reproduce low-frequency (multi-centennial) variability (see also *MacMynowski et al.* (2011); *Lovejoy et al.* (2013); *Geoffroy et al.* (2013); *Fredriksen and Rypdal* (2016)).

(II) It depends on the perfect power-law scaling of the structure functions of forcing and response, i.e., that these processes belong to the multifractal class (*Mandelbrot et al.*, 2008; *Rypdal and Rypdal*, 2016a). This is not true for e.g., the volcanic forcing (see Fig. 6c) nor for GCM responses (see Fig. 8).

(III) The analysis does not account for the internal variabiliy. The authors have argued that internal variability may be negligible compared to forced variability on the longest time scales. In Sect. 2.6 we demonstrated that this is not the case for GCMs. One should also keep in mind that for analysis of intermittency, the emphasis is on the smallest time scales. The intermittency of the temperature signal will be strongly influenced by, or even dominated by, the internal noise, and hence there is no reason there should be a strong similarity between intermittencies of forcing and temperature in a linear response model.

## 3.2 Effect of imperfect power laws on intermittencies

Here we present some theoretical considerations which demonstrate that imperfect scaling (power laws) of the response kernel and the structure functions can lead to different intermittency of forcing and response in a linear response model. In Sect. 3.3 we demonstrate this by an example, so the present subsection can be skipped by readers who are only interested in such a demonstration. The general, linear response model Eq. (1) can be written as a convolution of the forcing $F(t)$ with a response kernel $G(t)$;

$$\hat{L}[F(t)] = \int\limits_{-\infty}^{\infty} G(t-t')F(t')\,dt'. \tag{13}$$

For a general analysis of moments it is convenient to formulate the moments in the frequency domain rather than the time domain. Thus, we Fourier transform Eq. (13) to write

$$\mathcal{T}(f) = \mathcal{G}(f)\mathcal{F}(f), \tag{14}$$

where $\mathcal{T}(f)$, $\mathcal{F}(f)$, and $\mathcal{G}(f)$, are the Fourier transforms of $T(t)$, $F(t)$, and $G(t)$, respectively. By defining structure functions in frequency domain, $\mathcal{S}_q^T(f) \equiv \langle|\mathcal{T}(f)|^q\rangle$, $\mathcal{S}_q^F(f) \equiv \langle|\mathcal{F}(f)|^q\rangle$, we have the general, linear response model formulated as a linear relation between forcing and response structure functions of order $q$ in the frequency domain, with the ensemble average of the $q$'th power of the transfer function $|\mathcal{G}(f)|$ as a constant of proportionality;

$$\mathcal{S}_q^T(f) = \langle|\mathcal{G}(f)|^q\rangle \mathcal{S}_q^F(f). \tag{15}$$

L&V assume a power-law, linear response. This corresponds to a response function of the form

$$G(t) = \xi (t/\mu)^{H-1/2} \theta(t), \tag{16}$$

where $\xi = 1\,\mathrm{Km^2J^{-1}}$, $\mu$ is a constant in units of time which characterises the strength of the response, $H$ is the scaling exponent for the response used by L&V, and $\theta(t)$ is the unit step function. The Fourier transform of this response function yields (see *Rypdal and Rypdal* (2014))

$$|\mathcal{G}(f)| = \left(\frac{f}{f_0}\right)^{-(H+1/2)}, \tag{17}$$

where

$$f_0 = \frac{[\xi\mu\Gamma(H+1/2)]^{\frac{1}{H+1/2}}}{2\pi\mu},$$

and $\Gamma(x)$ is the Euler Gamma function. Hence the L&V special case of Eq. (15) is

$$\mathcal{S}_q^T(f) = \left(\frac{f}{f_0}\right)^{-q(H+1/2)} \mathcal{S}_q^F(f). \tag{18}$$

The next assumption made by L&V is that both forcing and response exhibit multifractal scaling. If we write the structure functions as (dropping the superscripts);

$$\mathcal{S}_q(f) = C_q(f)\, f^{-\eta(q)}, \tag{19}$$

the multifractal scaling assumption is that the multiplicative factor $C_q(f)$ is independent of the frequency $f$, such that the structure functions are perfect power laws in $f$ (*Mandelbrot et al.*, 2008). This is a very restrictive assumption that is not satisfied by any of the data in this study. If Eq. (19) holds true a plot of $\log \mathcal{S}_q(f)$ vs. $\log f$ is linear with slope $-\eta(q)$. The essence of the L&V approach (although some technicalities differ) corresponds to fitting the $\log \mathcal{S}_q(f)$ vs. $\log f$ curves with straight lines at the highest frequencies, or in other words, to draw tangent lines to the curves at the Nyquist frequency $f_N$. The negative slopes of these lines are interpreted as the scaling functions $\eta(q)$. This corresponds to defining the scaling functions by

$$\eta(q) = \left[\frac{dS_q(f)}{d(\log f)}\right]_{f=f_N}, \tag{20}$$

and from Eq. (19) we then find the $f$-dependence of $C_q(f)$ which represents the deviation from multifractal scaling. The L&V approach includes normalizing the signals $T(t)$ and $F(t)$ such that they have the same power at the lowest frequency $f = 1$, i.e., $S_2^T(1) = S_2^F(1)$. If $H \neq -1/2$ Eq. (18) then implies that $f_0 = 1$, and putting $f = 1$ in Eqs. (18) and (19) we find,

$$\mathcal{S}_q^T(1) = \mathcal{S}_q^F(1) = C_q^T(1) = C_q^F(1) \tag{21}$$

for all all $q$. From the logarithm of Eqs. (18) and (19) we find for $f > 1$,

$$\eta_F(q) - \eta_T(q) + q(H+1/2) = \frac{\log[C_q^F(f))/C_q^T(f)]}{\log f}. \tag{22}$$

If $T(t)$ and $F(t)$ exhibit perfect multifractal scaling we have $C_q(f) = C_q(1)$, and from Eq. (21) the right hand side of Eq. (22) vanishes. Hence, for this case we have the L&V results that the curves $\eta^T(q)$ and $\eta^F(q)$ have the same curvature, i.e., the response and forcing exhibit the same multifractal intermittency. However, the term $q(H + 1/2)$ on the left hand side arises from the particular power-law form of the linear response function shown in Eq. (17). With another form of the linear response kernel this term might not be linear in $q$, and this could introduce different curvature of $\eta^T(q)$ and $\eta^F(q)$. Different curvature is also introduced if the structure functions are not perfect power laws. Then the term on the right of Eq. (22) will in general not vanish, and it may have a non-zero second derivative. This may give rise to different curvatures of $\eta^T(q)$ and $\eta^F(q)$ even if the response is linear with the power-law response kernel given by Eq. (17).

### 3.3 Response to volcanic forcing

An important point in L&V is that intermittency in volcanic forcing and the corresponding temperature response are different, and that this is a signature of nonlinearity in the response. In this subsection we shall first demonstrate that the intermittency in the volcanic forcing is not multifractal, i.e., all the structure functions are not power laws. This is a symptom of the lack of correlations between bursts that characterises a multiplicative cascade. Next, we shall show by using L&V's trace moment analysis on a simple, linear response model, that we can reproduce the intermittency observed in the response to volcanic forcing in the ZC model. This linear response exhibits a similar power spectrum, similar trace moments, and almost identical intermittency parameters as the ZC response. And more important; these features are considerably different in the forcing and the response, even though the response model is linear. It demonstrates that these results obtained from the ZC model is not a signature of nonlinearity in the response.

Let us first build some intuition on the nature of the volcanic forcing. In Fig. 6a we have zoomed in on the volcanic forcing signal used in the ZC model. Each volcanic eruption is represented by 2-3 data points (years) different from zero (some large eruptions are represented by a few more points). If the eruptions are distributed randomly in time (Poisson distributed) the autocorrelation function (ACF) will vanish after a time lag of a few years. This is exactly what we observe in Fig. 6b. The spectral structure functions used in Sect. 3.2 are convenient for theoretical studies, but not for estimation based on short and spiky time series. Here it is better to use the standard structure functions which are computed from the empirical moments;

$$\hat{S}_q(\Delta t) = (N - \Delta t)^{-1} \sum_{t=1}^{N-\Delta t} |Y(t + \Delta t) - Y(t)|^q \tag{23}$$

where $Y(t) = \sum_{t'=0}^{t} F(t')$ is the cumulative sum of the forcing time series. This is a standard estimator commonly used in analysis of stationary time series. It is much more transparent than the trace moments employed by L&V, and contains no hidden assumptions about power-law structure functions or the existence of an "outer scale" for these power-laws (see discussion in Sect. 4.2). The

empirical moments of the volcanig forcing signal are shown in Fig. 6c. The steeper slopes (slope $\approx q$) for $\Delta t \leq 4$ is due to the smoothness of the forcing signal on these short time scales, signified

by the ACF in Fig. 6b. For $q = 2$ the structure function looks quite straight and with slope close to 1 in the log-log plot for the scale range 4-100 yr. For smaller $q$ the plots become more curved. This is symptomatic for a stationary, uncorrelated process (Lévy process) which is non-Gaussian on short time scales, although the central limit theorem requires that it converges to a Gaussian on the longer scales. According to *Mandelbrot et al.* (2008), such a process is not multifractal (see also Sect. 2.5

and appendixes in *Rypdal and Rypdal* (2016a)). In practice, L&V's approach corresponds to assuming that the moments can be written in the power-law form $\hat{S}_q(\Delta t) \sim \Delta t^{\zeta(q)}$, where the scaling function $\zeta(q)$ is estimated by fitting straight lines to the structure functions in the log-log plot in the range 4-100 yr. This has been done in Fig. 6d. The curved scaling function is interpreted by L&V as a signature of multifractality, but this interpretation is correct only if all structure functions are power

laws (straight lines in log-log plots). It is easily demonstrated that very similar results are obtained by random shuffling of the onset times of the volcanic spikes, which would convert a multifractal signal into a realisation of a Lévy process. If the original signal were a multifractal, the result should be quite different after shuffling. For a deeper discussion of these disagreements see the interactive discussion and in particular our author comment AC3 (*Rypdal and Rypdal*, 2016b).

Our main focus here, however, is not on the incorrect multifractal interpretation of the scaling analysis, but on the incorrect conclusions drawn from this analysis when it comes to nonlinearity in the response. As a means to investigate this point we construct a linear response model that mimics the ZC response to the volcanic forcing. The ZC response is shown by the blue curve in Fig. 7a. We observe that every volcanic spike seems to be succeeded by a damped oscillation. Thus, we construct

a linear, damped harmonic oscillator response model and select the parameters to produce a response signal that looks similar to that of the ZC response to the volcanic forcing when we drive the model with stochastic forcing in addition to the volcanic forcing. We make no attempts to fine-tune the model parameters, since this extremely simple model obviously is not an accurate substitute for the ZC model. The purpose of devising this model is only to demonstrate that a linear model can produce

a response with intermittency parameters very different from those of the forcing. These are results which L&V contend can only arise from nonlinearity of the response.

The response according to the linear model is shown by the red curve in Fig. 7a, and we compute the trace moments and intermittency coefficients for this linear response signal. We have used the Mathematica routines downloaded from Shaun Lovejoy's web page for these computations to make

sure that the results are comparable to those presented by L&V. Fig. 7b is a reproduction of Figure 6a, top right, in L&V for the volcanic forcing. L&V interpret the wide spread in slopes of the trace-moment curves as signature of multifractal intermittency, and they compute the intermittency coefficients $C_1 = 0.48$ and $\alpha = 0.31$ (their Table 1). The results depend on the exact fitting range chosen, so we cannot expect to get exactly the same results for these parameters. We find $C_1 = 0.52$

and $\alpha = 0.13$ (which makes us wonder if $\alpha = 0.31$ in L&V is a misprint). In Fig. 7c we have computed the trace moments for an arbitrary realisation of the linear response model. This figure is very similar to their Figure 6a, bottom right, for the ZC response. The intermittency parameters computed by L&V for this case are $C_1 = 0.054$ and $\alpha = 2.0$, while our results for the linear model are $C_1 = 0.039 \pm 0.013$ and $\alpha = 1.92 \pm 0.03$. These numbers are mean values over an ensemble of 100

realisations of the linear response model and the errors are $\pm 2\sigma$, where $\sigma$ is the standard deviation over this ensemble. The important feature here is not the similarity between the intermittency parameters for the ZC model and this linear model, but rather the great difference in these parameters between volcanic forcing and response in the linear model. L&V interpret this difference as a signature of nonlinearity, but our exercise shows that such a difference can be obtained from a simple

linear response model with internal noise.

### 3.4 Intermittency in GCMs

The breakdown of condition (III) due to internal variability in GCMs is clearly illustrated in Fig. 8, which is based on the data from the NorESM model. Fig. 8a shows the volcanic forcing signal (red), and the model response to this signal (black). Fig. 8b shows a signal composed of two components;

one is the volcanic forcing signal normalised such that the magnitudes of the large volcanic spikes roughly match those of the volcanic response signal. This signal can be thought of as the instantaneous response to the stochastic forcing. The other component is the internal variability represented by a control run. This composite signal represents a trivial linear transformation (multiplication by a normalization factor) plus a signal representative for the internal variability. Figs. 8c,d show the

structure functions (SFs) and the scaling function for the volcanic forcing computed from straight lines fitted to the SFs in the range displayed in Fig. 8c. Figs. 8e,f show the same for the model response signal to the volcanic forcing, and Figs. 8g,h for the the composite signal shown in Fig. 8b. According to L&V (who believe condition III is irrelevant), the intermittency shown by the curvature of the scaling function in Figs. 8d should be preserved in the scaling function for the composite

signal shown in Fig. 8h, but it is not. The latter signal is almost non-intermittent due to the "contamination" from the internal noise. The contamination explains the reduced intermittency observed in the response to the volcano forcing shown in Figs. 8e,f. This proves that nonlinearity in the response is not required to explain the difference in intermittency between forcing and response in GCMs.

### 4 Discussion

In this section we summarise our view on some of the relevant issues raised by L&V in the interactive discussion. The readers are referred to the interactive discussion for details.

### 4.1 Physics frameworks

A qualitative mental picture, a theoretical framework or paradigm, of the physical mechanisms that govern the macro dynamics is of course an important guideline when formulating and testing hypotheses. But it can also become a straitjacket that restricts the range of alternatives that one is willing to investigate. In the interactive discussion (comment SC3), *Lovejoy et al.* (2016) write:

> ... at short enough time scales, when external forcings are small enough, then theoretically we may expect the atmospheric response to be approximately linear, however, at long enough time scales, due to temperature-albedo feedbacks, the response is expected to become nonlinear. At the same time, it is possible that at long enough time scales, due to quite different surface and atmospheric interactions, that solar and volcanic external forcings combine nonlinearly.

Our comment to this is that we find no reason why responses should be more linear on short than on long time scales, in particular not the response to the burst-like volcanic forcing. The response of local climatic variables on synoptic and seasonal scales to strong volcanic eruptions is certainly nonlinear. But on longer time scales, the global temperature will change in proportion to the change in heat content in the upper ocean, which again will change in proportion to the net radiative flux. The response in presence of feedbacks that modify the radiative flux is not generally expected to become nonlinear. Feedbacks are typically modelled linearly, although in some cases different feedbacks may combine nonlinearly.

The ENSO phenomenon is probably a nonlinear mode in the climate system, and is part of the internal variability, even though it can be influenced by external forcing. The nonlinear nature of the oscillation makes it likely that the timing of El Niño events can be influenced by external forcing such as strong volcanic eruptions. In general, the modes of internal variability of the climate system are results of nonlinear processes, and the modes are probably responding nonlinearly to external forcing. But we find it less likely that the ensemble averaged, global temperature response is nonlinear to an extent that is detectable.

The trace moment analysis applied by L&V is rooted in ideas of intermittency and multifractality, which have emerged from turbulence theory. It was used by *Schertzer and Lovejoy* (1987) in the context of rain and cloud fields, but more recently they have worked extensively to extend the ideas of turbulent, multiplicative cascades, not only to atmospheric dynamics and weather, but also to climate dynamics across a vast range of scales (Lovejoy and Schertzer, 2013; Lovejoy, 2014). The validity of extending of the turbulence framework to encompass the dynamics of the entire climate system across the scales is not obvious and deserves to be challenged. The simple, linear energy-balance modelling is one example of an alternative framework. Models of this kind can be extended to incorporate several interacting subsystems with different response times (multi-box models), and can give rise to responses that are close to power laws over a certain range of scales (see Fig. 5). But there are

also other competing paradigms based on treating the climate as a high-dimensional dynamical system residing in non-equilibrium stationary states, and invoking response theory of non-equilibrium statistical mechanics for prediction of changes in the globally averaged surface temperature as well as its spatial patterns (see *Lucarini et al.* (2016) and references therein). For the foreseeable future, climate science will probably have to live with different and complementary theoretical frameworks for understanding and predicting the future climate state under variable forcing. This is important to keep in mind when formulating and testing hypotheses.

## 4.2 Framework-biased tests

Tests formulated on the basis of one particular theoretical framework runs the risk of becoming self-fulfilling. The trace moment analysis employed by L&V explicitly assumes the existence of multifractal scaling up to a certain "outer scale," and lines are fitted to the trace moments under the constraint that they all cross at this outer scale. The slopes of these lines are used to compute the intermittency parameters, even in cases where these lines are poor fits to the actual trace moments. The method is automatised and contains no means to discriminate between true multifractal and non-Gaussian uncorrelated processes (Lévy processes). The interactive discussion has revealed that L&V do not distinguish between these two classes of processes. The implication of failing to make this distinction is that a mathematical result for multifractal processes (stating that a linear transformation preserves intermittency) is applied by L&V to processes for which this result is not valid.

Application of estimators and tests to signals for which they are not designed is very common in scaling analysis, and often leads to meaningless results which depend on the selected estimator and on the range of scales on which the estimation is based. The most common are estimators for scaling exponents of mono-scaling processes applied to signals that are neither mono- nor multi-scaling. The trace moment test is more subtle than these, but the nature of the error is the same. The test is based on a certain scaling paradigm, and does not test whether the signal actually satisfies those scaling properties.

### 4.3 How to avoid positive-outcome bias?

The approach in L&V's paper is to argue forcefully in favour of a preconceived position, and to present carefully selected evidence in favour of it. This is a classical prescription for production of false positive results. Our comment does not prove that their conclusions are false (nonlinearity cannot be falsified), but it points out that their tests have not detected nonlinearity.

Recent bibliometric studies show that there is a clear trend towards an increasing positive-outcome bias in scientific publications. *Fanelli* (2012) analysed over 4600 papers published in all disciplines between 1990 and 2007, and found that the proportion of negative results dropped from 30% to 14% in this period. The decreasing number of negative results reported is linked to an increasing number of false positive results published throughout the scientific literature (*Ioannidis*, 2005).

We would use this opportunity to advocate a practice of clearer distinction between different kinds of scientific activities and how results from these activities are disseminated. An essential part
of scientific endeavour is the inspired idea that leads to proposition of a bold hypothesis. Another is the purely deductive process, which encompasses mathematics. These two constitute what we could classify as theory. A third activity is the collection of empirical information, and a fourth is the testing of hypotheses against the empirical data. The creative process leading to formulation of new hypotheses should not be subject to censorship, and we see few problems of publishing "wild ideas"
as long as they are presented as such. A problem arises, however, when the inspired and uncritical narrative leading up to a bold idea is blended with selected empirical evidence and/or tests that are designed to confirm, rather that to reject.

## 5 Conclusions

A correct treatment, without unjustified approximations, of the issue of additivity in the Zebiak-Cane
model gives no reason for rejection of a linear response model (see Fig. 2). This conclusion holds even without accounting for internal variability, but is enforced by the inclusion of this effect. This was demonstrated in Sect. 2.6 by the alternative test introduced in Sect. 2.4.

L&V's analysis of intermittencies is based on a mathematical result which states that if the response is linear the intermittency computed through trace moment analysis must be the same in
forcing and response. However, this result holds only if both forcing and response belong to the class of multifractals, i.e., if all structure functions are power-laws (*Mandelbrot et al.*, 2008), and in addition it requires that the response function is a power law in the entire scale range of interest. Fig. 6 demonstrates the structure functions of volcano forcing are not power laws, and Fig. 8 that this is the case also for structure functions of global temperature in GCMs. Fig. 5 shows that the
temperature response function in GCMs is not a power-law on all available time scales.

In Fig. 7 we illustrated by an example that the intermittencies can be very different in forcing and response produced by a linear response model with internal variability. The rôle of internal variability in reducing the intermittency in the linear response to an intermittent forcing was demonstrated for GCM data in Sect. 3.4 and displayed in Fig. 8. Hence, our conclusion is that the intermittency
analysis of L&V does not constitute a valid test for rejecting the linear response hypothesis.

*Acknowledgements.* This paper was supported by the the Norwegian Research Council (KLIMAFORSK programme) under grant no. 229754. We are grateful to Shaun Lovejoy for providing Mathematica codes for estimation of Haar structure functions and trace moments on his web page: http://www.physics.mcgill.ca/ gang/software/index.html, and to Hege-Beate Fredriksen for debugging these codes.

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

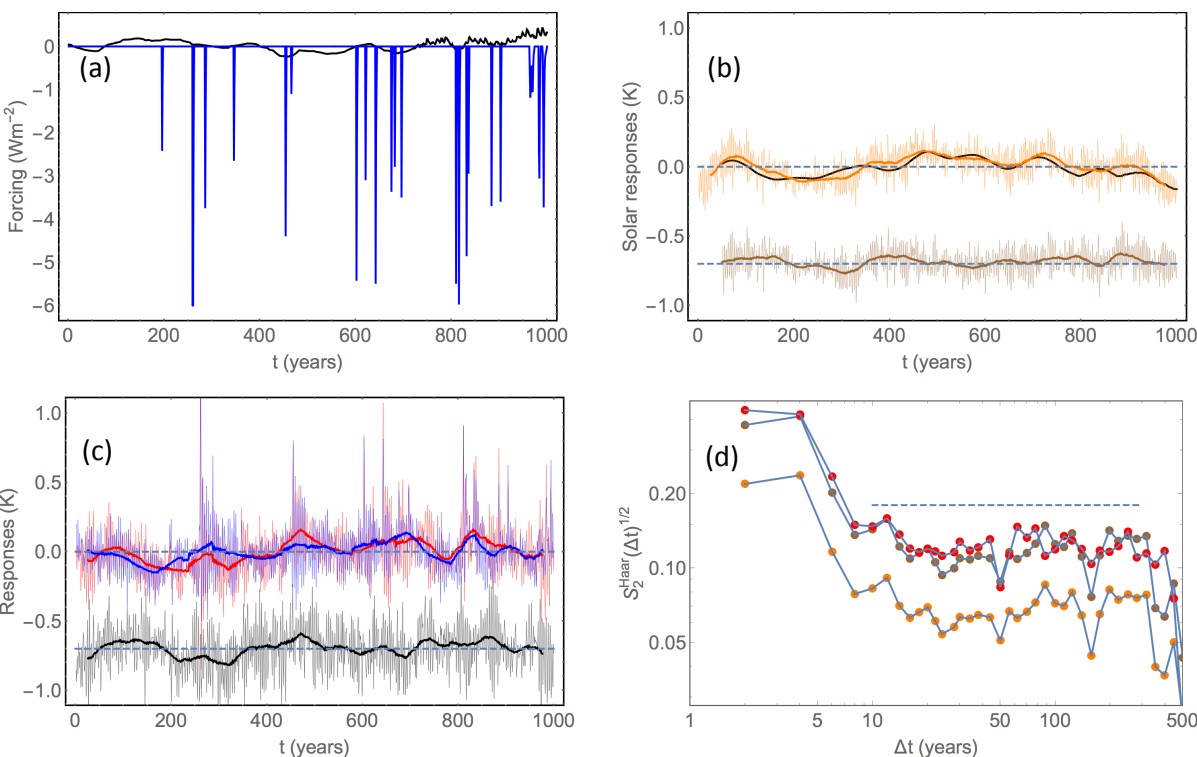

**Figure 1.** (a): Time series of the solar (black) and volcanic forcing (blue) for the last millennium used in the simulations of the ZC model. (b): Responses after averaging over 100 realisations. Thin, orange curve is response to solar forcing and the thick, orange is filtered by a 50 yr moving average. The thick, black curve is the filtered and shifted solar forcing signal $\Delta T_s^{\text{det}}(t, \Delta t)$ given by Eq. (8). The brown, thin curve is the internal noise $\epsilon(t)$ defined in Eq. (9), and the thick brown is the filtered time series. (c): Thin, blue curve represents $T_{s+v}(t)$, the thin, red curve is $T_s(t) + T_v(t)$, and the thin, black curve is their difference $\varepsilon(t) = T_s(t) + T_v(t) - T_{u+v}$. Thick curves are the corresponding filtered series. (d): Haar structure function of $\epsilon(t)$ (orange bullets), of $\varepsilon(t)$ (red bullets), and of $\sqrt{3}\epsilon(t)$ (brown bullets).

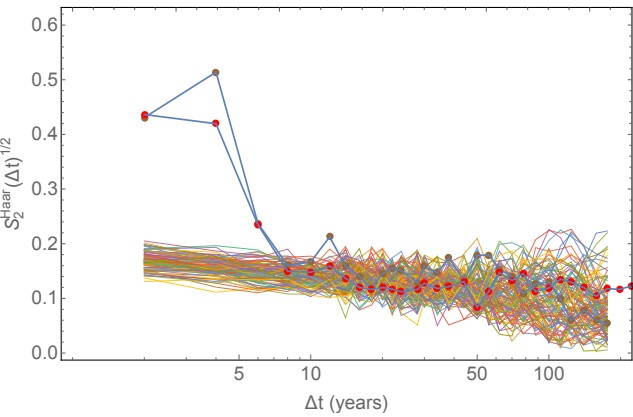

**Figure 2.** Brown bullets: Haar fluctuation function of $\sqrt{3}\epsilon(t)$, where $\epsilon(t)$ is the first 195 yr of the volcanic forcing record. Red bullets are Haar-fluctuations of $\varepsilon(t)$. These two curves look similar to the corresponding curves in *Lovejoy et al.* (2016). The crucial issue is whether the difference between these two curves is statistically significant. The thin curves constitute Haar fluctuations of a 100 member ensemble of fractional Gaussian noises (fGn's) of 195 yr length with $H = -0.1$ ($\beta = 2H + 1 = 0.8$). On time scale less than 10 yr the fGn is not a good model for the internal noise because of the ENSO dynamics, but on longer time scales the flat Haar-fluctuation curve suggests that an fGn with $\beta \approx 0.8$ is a crude statistical model of the internal variability. The scatter of the Haar fluctuation in this ensemble gives an idea about the statistical uncertainty of an estimate of internal variability based on a 195 yr long record. This uncertainty exceeds the estimate of $|\sqrt{3}\epsilon(t) - \varepsilon(t)|$ (the difference between the brown and the red curves), hence this difference is not statistically significant.

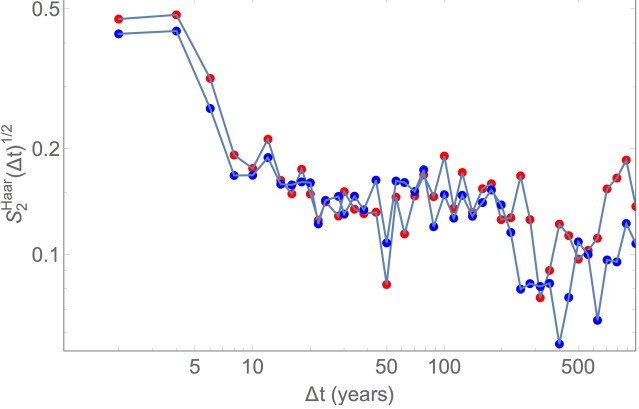

**Figure 3.** Haar structure functions $\sqrt{\langle|\Delta T_s + \Delta T_v|^2\rangle}$ (red bullets) and $\langle|\Delta T_{s+v}|^2\rangle$ (blue bullets).

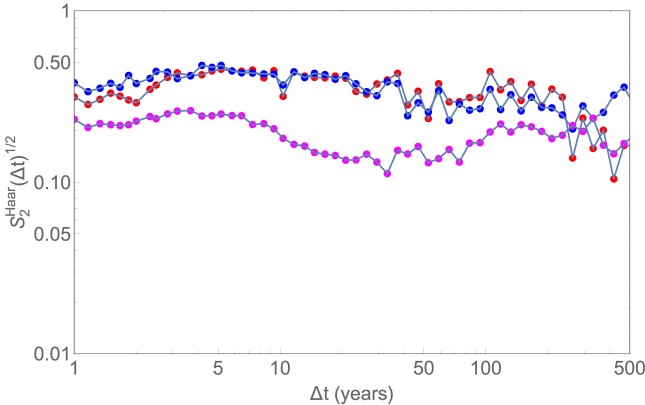

**Figure 4.** Haar fluctuations for NorESM data. Red curve: Haar fluctuation of the response to solar + volcanic forcing. Blue curve: the Haar fluctuation of the summed solar and volcanic response. Magenta curve: Haar fluctuation of the control run.

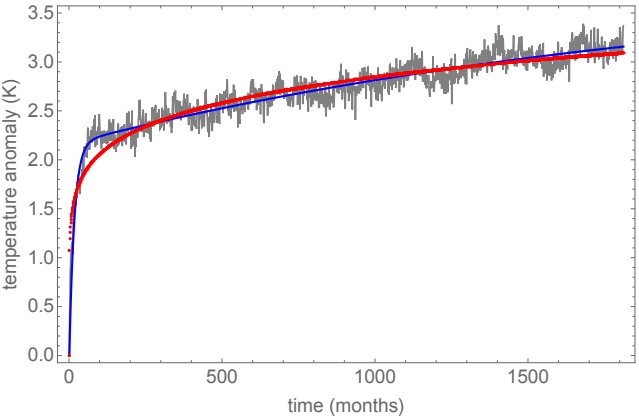

**Figure 5.** Grey curve is the global temperature response to a sudden 4-doubling of atmospheric $CO_2$ concentration in the GISS-E2-R model. Blue curve is a fit of superposition of two exponential responses (two-box model solutions); the two exponential time constants being $\tau_1 = 1.3$ yr and $\tau_2 = 176$ yr. Red curve is a power-law fit, and is a poor fit up to several years.

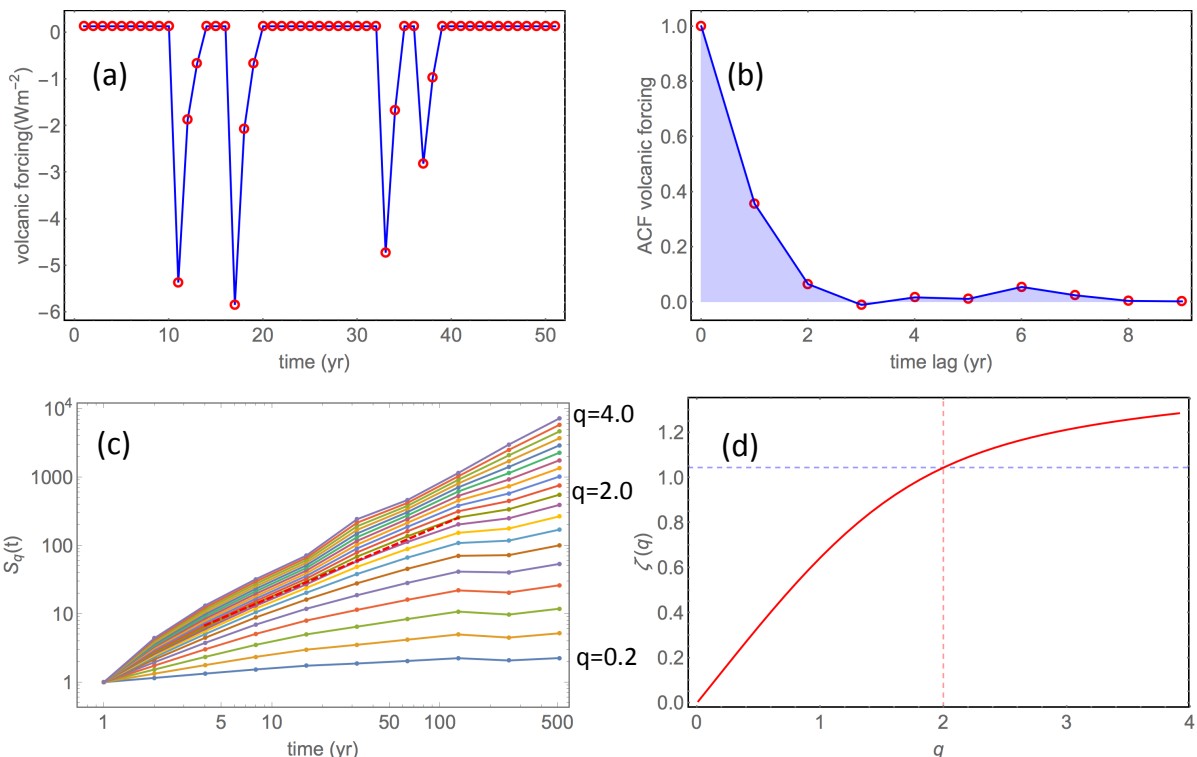

**Figure 6.** (a): A zoom-in on the volcanic forcing signal shown in Fig. 1a. (b): The ACF estimated for the volcanic forcing signal. (c): The structure functions (empirical moments) $\hat{S}_q(\Delta t)$ for the volcanic forcing signal estimated for $q = (0.2, 0.4, \ldots, 4.0)$ The red, dashed line is a linear fit to the log-log plot of $\hat{S}_2(\Delta t)$. (d): The scaling function $\zeta(q)$ computed from linear fits to the $\hat{S}_q(\Delta t)$'s over the interval $\Delta t \in (4, 128)$. The observation that $\zeta(2) \approx 1$ suggests that the process is uncorrelated on these time scales.

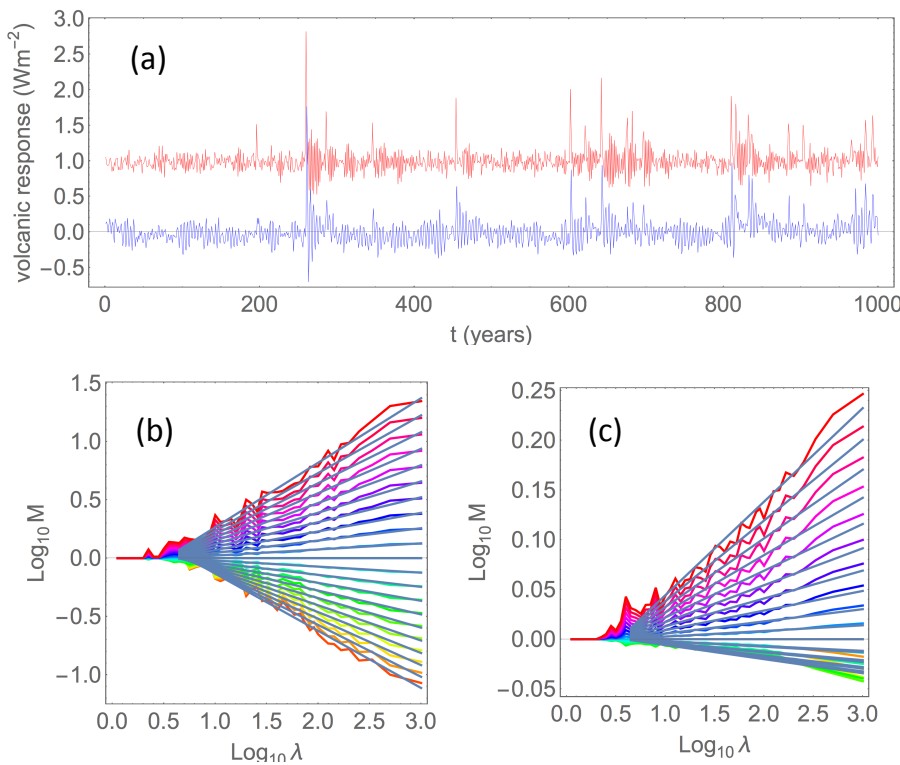

**Figure 7.** (a): Blue curve is the average over 100 realisation of the response to volcanic forcing in the ZC model, and the red curve is the response to this forcing plus a stochastic Gaussian white noise forcing in a linear, damped harmonic oscillator model. (b): Result of trace moment analysis of the volcanic forcing signal. It is very similar to the corresponding panel in Figure 6 of L&V. (c): Result of trace moment analysis of the harmonic oscillator response shown by the red curve in panel (a). It is very similar to the corresponding panel for the ZC response to volcanic forcing in Figure 6 of L&V.

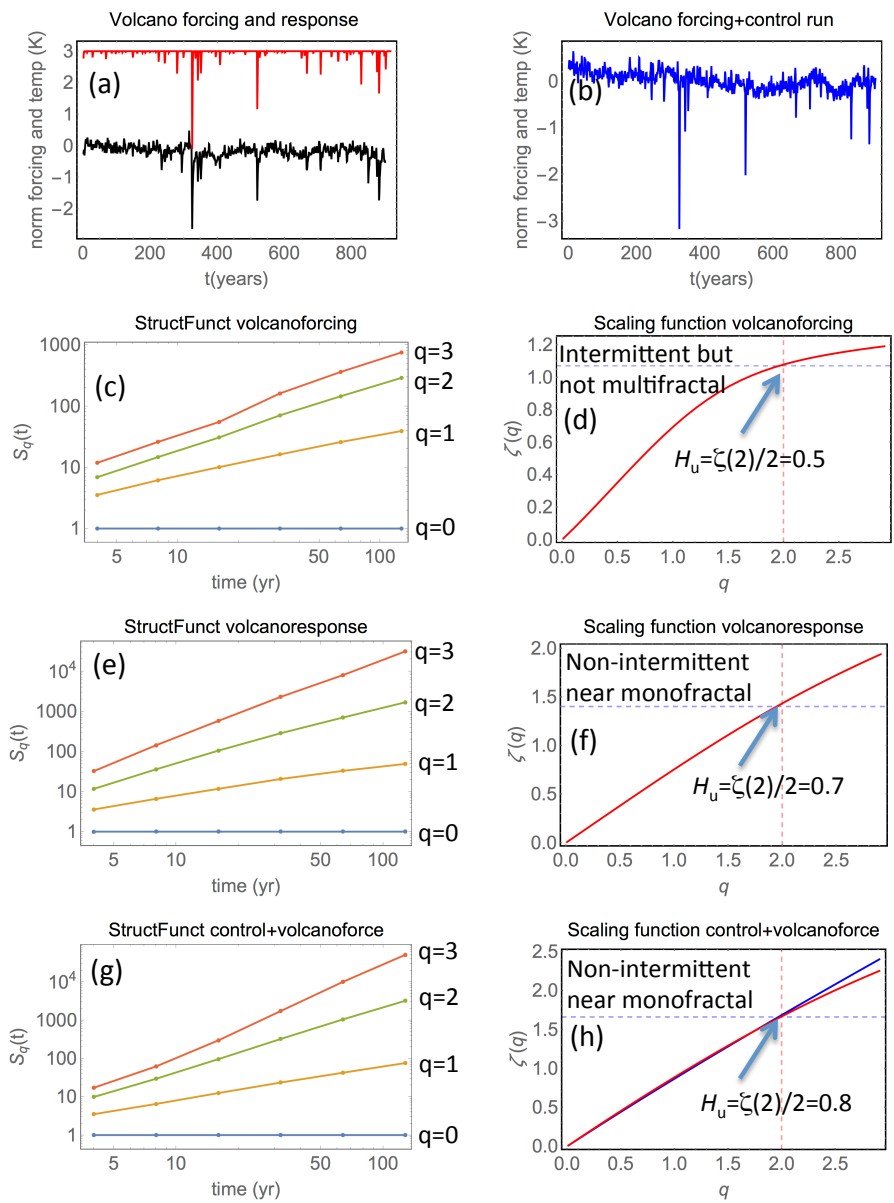

**Figure 8.** Analysis of global temperature responses in the NorESM model. (a): the volcanic forcing (red) normalized such that the larges spikes are approximately equal to the spikes of the response signal (black). (b): the red curve in (a)+the control-run temperature signal. (c): structure functions (of cumulative sum) of volcano forcing. (d): scaling function derived from (c). (e): structure functions of the response to volcanic forcing. (f) scaling functions derived from (e). (g): structure functions of the signal in (b). (h): scaling function derived from (g). The red line arise from fitting straight lines in the entire scale range plotted, 4-128 yr. The blue line is from fitting only in the scale range 16-128 yr. It shows weak intermittency in both cases, but also that estimated intermittency depends on the scale range chosen for fitting. The difference in curvature (reduction of intermittency) between (d) and (h) is exclusively caused by the addition of the internal noise represented by the control run, and the similarity between (f) and (h) indicates that the internal variability is the main cause of reduced intermittency in the response to volcanic forcing.