# Peer review of "Comment on "Scaling regimes and linear/nonlinear responses of last millennium climate to volcanic and solar forcing" by S. Lovejoy and C. Varotsos"

_Earth System Dynamics, 2016_

## Short Comment (SC1) · 3 Apr 2016

**On the importance and significance of Intermittency**
**Rebuttal of Section 3 of Rypdal and Rypdal 2016**
S. Lovejoy, C. Varotsos

**Intermittency and volcanism**

Although this was originally motivated as a response to Rypdal and Rypdal 2016 (R+R below) we well this as an opportunity to clarify some important points about intermittency in general and volcanic intermittency in particular. Since intermittency is not well known outside the turbulence community, these clarifications could be of wide interest to the community. The response to the first part of R+R will be given in a separate comment.

The nonlinearity of the atmospheric response to strong, "spikey", intermittent forcings such as volcanism was noticed over twenty years ago by [*Clement et al.*, 1996] who found that in numerical climate models, there is a high sensitivity to small forcings and a low sensitivity to large forcings. Nonlinearity response to volcanism was also claimed by [*Mann et al.*, 2005], indeed, it was a main purpose of his simulations of the interaction between El Nino and volcanism that were analysed in [*Lovejoy and Varotsos*, 2016] (henceforth L+V). Therefore, the primary contribution of the corresponding section 4 in L+V is to quantify this. The L+V method is straightforward, it simply exploits the known fact that linear transformations of a time series can only make linear changes to the exponent scaling function $\xi(q)$, they cannot affect the nonlinear part that is associated with the intermittency (see e.g. ch. 5 of [*Lovejoy and Schertzer*, 2013]).

This simple and fundamental result has been known since at least the 1980's and was explained in the original L+V text; it is correct. Although R+R complain that L+V paper had long-winded and unnecessary reviews of theory, they would be advised to look more carefully at the discussion of the trace moment analysis that quantified the intermittency in section 4 of L+V – or better still – to consult chapters 3-5 of [*Lovejoy and Schertzer*, 2013]. Trace moment analysis was originally developed thirty years ago as a sensitive way to quantify multifractal intermittency in turbulence. The very first step in the analysis is precisely a nonlinear transformation of the series so as to obtain an estimate of the underlying driving fluxes: the absolute values of the first or second differences are commonly used as flux estimates. The reason that trace moment analysis is so effective is that it removes the linear $qH$ term in the structure function exponent so that the nonlinear $K(q)$ part can be studied directly. Had R+R noticed this fact, they would not have bothered to develop the linear analysis (eqs. 13 to 22) which is irrelevant to the trace moment analyses presented in L+V and to its conclusions.

But even if the trace moment analysis had been a linear one, the R+R analysis would still be of little interest since it makes unnecessarily restrictive assumptions (and incorrectly imputes them to L+V)! For example R+R's affirmation II, that an assumption of multifractality is needed: "i.e., that these processes belong to the multifractal class" is not true. It is enough to recall that for a scaling process it is enough that the *dominant statistics* vary as power laws with scale, and that this is usually accompanied by all kinds of sub-exponential corrections and these are

missing in eqs. 15- 22.

Interestingly - although hardly surprisingly – volcanism is an excellent example of a strongly intermittent multifractal process. This fact was first pointed out in [*Lovejoy and Schertzer*, 2012] and [*Lovejoy*, 2014] used the estimated multifractal exponents to produce the highly realistic multifractal simulations reproduced fig. 1a which includes one real series. Can the reader spot the fakes?

R+R complain that because the volcanic forcing series may be approximated by a distribution of Levy type (power law) spikes, (each more or less uniformly distributed along the time axis) that this somehow contradicts the multifractality of the process. This is a misunderstanding. Although cascade processes are the generic multifractal process, there are many, many ways of generating a multifractal process and their proposal to distribute Levy eruptions uniformly along the time axis is indeed one such a proposal (note that for more generality, these could be distributed over a fractal set). No matter which model is closer to reality - the Levy spikes – or a cascade process, they are both strongly intermittent, multifractal and the L+V conclusions hold. In any case, fig. 1a shows that cascades can approximate volcanic processes quite well. And the L+V theorem that linear filters can only make linear changes in structure function exponents – and hence cannot change the intermittency - remains valid.

Just to complete the picture – to demonstrate without using trace moments - the strong differences in multifractal intermittency between the volcanic forcings and the responses, we have added fig. 1b which shows the ratio of the first order moment to the RMS moment of the Haar fluctuations. A nonintermittent, quasi-Gaussian process would be completely flat; nonzero slopes are consequences of intermittency. The reference slopes quantify the intermittency (for log-normal multifractals, the slope is the codimension of the mean, $C_1$ but at least for the volcanic forcings, this is a poor approximation). The values indicated (0.05, 0.30) are very different: the former is roughly the typical value of the intermittency of wind in atmospheric turbulence whereas the latter value corresponds to precipitation (see Box 4.1 "Overview of the horizontal scaling properties of atmospheric fields" in [*Lovejoy and Schertzer*, 2013]).

Finally, the fact that adding a white noise to the ZC volcanic response can lead to an apparently low intermittency multifractal process (R+R's fig. 4c) only shows that when the intermittency is low and the range of scales is not so large, care must be taken in interpreting the results (they could see a useful discussion of this in appendix 4A in [*Lovejoy and Schertzer*, 2013], or the curved quasi-Gaussian envelopes in fig. 6c). In any case there is a huge difference in the intermittency between R+R's fig. 4b and 4c (the volcanic forcing and response), just look at the range of the fluctuations at the large scale ratios (the small resolutions) at the right of the diagrammes (a factor of about 30)!

**Erronneous Structure functions analyses:**

It is unfortunate that the R+R comment was marred not only by an inappropriately harsh tone, but also by some real difficulties with the data analysis. We would not dwell on it except that it demonstrates some of the pitfalls of scaling analyses, and it may therefore be of general interest.

Consider R+R's treatment of the volcanic forcings, their Auto Correlation Function (ACF) and structure function analyses (fig. 3c, d). First, the decrease of the ACF with scale (fig. 3b) proves little. However, if the ACF decreases, then – at least for stationary processes (and their structure function analysis eq. 23 assumes this), the structure functions must simply asymptote to a maximum value: in obvious notation: $\left\langle \Delta F(\Delta t)^2_{diff} \right\rangle = 2\left(\left\langle F(t)^2 \right\rangle - \left\langle F(t)F(t-\Delta t) \right\rangle\right)$. From this standard equation, it is obvious that the decrease of the ACP ($\left\langle F(t)F(t-\Delta t)\right\rangle$) at large $\Delta t$ leads the difference structure function ($\left\langle \Delta F(\Delta t)^2_{diff} \right\rangle$ to asymptote to the series variance ($\left\langle F(t)^2 \right\rangle$) which (in this case) is determined by the high frequency details of the series. That this is indeed the relevant explanation is shown in the (correct) difference based structure function analysis in fig. 2a below. Somehow, R+R must have made an error since their fig. 3c shows an impossibly increasing difference structure function. Indeed – as pointed out in the recapitulative part of L+V – when $H$<0, one needs to define the fluctuations differently, Haar fluctuations being a convenient method (see the details in L+V). Fig. 2b below compares the Haar fluctuations with the appropriately modified DFA method, both of which confirm the analysis of L+V and contradict R+R's fig. 3c, d. Just to make things perfectly clear, we have added a spectral analysis (fig. 2b) that again confirms that the second order structure function exponent ξ(2) ≈ -0.90 (spectral exponent β = 1+ξ (2)=0.1) which is quite contrary to R+R who claim ξ(2) ≈ 1 and hence β =2, (they even suggest that this Brownian motion value has fundamental significance!). In all analyses, the R+R exponent is very far from the observations.

Although over thirty years ago, multifractal intermittency was a fundamental breakthrough in turbulence theory, its understanding and importance are not sufficiently appreciated outside the turbulence community. We would therefore like to thank R+R for giving us this opportunity to clarify this important question for the benefit of climate scientists.

[Figure]

Fig. 1a: This figure is reproduced from [*Lovejoy*, 2014]. It shows three "fake" volcanic reconstructions produced by highly intermittent multifractal simulations as well based on the [*Gao et al.*, 2008] volcanic reconstruction. The fakes are normalized as to have the same mean forcing (the absolute forcings are shown). To find which is the real series, the reader may consult the original paper.

[Figure]

Fig. 1b: This shows the ratios R of the first order (q=1) fluctuations with respect to the root mean square (RMS) fluctuations for a series $F(t)$. Top is for $F$ = the ZC temperature response to the volcanic response, the bottom is for $F$ = the volcanic forcing. A nonintermittent, quasi-Gaussian process would be completely flat, nonzero slopes are consequences of intermittency.

[Figure]

Fig. 2a: The first and second order structure functions for all intervals showing that they do roughly indeed asymptote to a constant as expected since H<0. Compare this to R+R's fig. 3c.

[Figure]

Fig. 2b: RMS Haar fluctuation analysis and the DFA scale function divided by $\Delta t$ (so as to correspond to the series rather than its integral/ running sum) and shifted in the vertical by a factor $\approx 70$. The solid reference slope corresponding to $\xi(2) = -0.90$ is shown for reference as well as the dashed reference corresponding to $\xi(2)=1$ (the value from fig. 3d in R+R).

[Figure]

Fig. 3: The actual volcanic forcing spectrum (blue) compared to the slope (solid reference line) inferred from the Haar and DFA analysis (fig. 2b) as well as the inferred slope from R+R's structure function, their fig. 3d (dashed).

**References:**

Clement, A. C., R. Seager, M. A. Cane, and S. E. Zebiak (1996), An ocean dynamical thermostat, *J. Climate*, *9*, 2190–2196.

Gao, C. G., A. Robock, and C. Ammann (2008), Volcanic forcing of climate over the past 1500 years: and improved ice core-based index for climate models, *J. Geophys. Res.*, *113*, D23111 doi: 10.1029/2008JD010239.

Lovejoy, S. (2014), Scaling fluctuation analysis and statistical hypothesis testing of anthropogenic warming, *Climate Dynamics*, *42*, 2339-2351 doi: 10.1007/s00382-014-2128-2.

Lovejoy, S., and D. Schertzer (2012), Stochastic and scaling climate sensitivities: solar, volcanic and orbital forcings, *Geophys. Res. Lett.*, *39*, L11702 doi: doi:10.1029/2012GL051871.

Lovejoy, S., and D. Schertzer (2013), *The Weather and Climate: Emergent Laws and Multifractal Cascades*, 496 pp., Cambridge University Press, Cambridge.

Lovejoy, S., and C. Varotsos (2016), Scaling regimes and linear/nonlinear responses of last millennium climate to volcanic and solar forcings, *Earth Syst. Dynam.*, *7*, 1–18 doi: doi:10.5194/esd-7-1-2016.

Mann, M. E., M. A. Cane, S. E. Zebiak, and A. Clement (2005), Volcanic and solar forcing of the tropical pacific over the past 1000 years, *J. Clim.*, *18*, 447-456.

---

## Author Comment (AC1) · 7 Apr 2016

Manuscript prepared for Earth Syst. Dynam. with version 2015/04/24 7.83 Copernicus papers of the LATEX class copernicus.cls. Date: 7 April 2016

**Reply to Lovejoy and Varotsos - I**

K. Rypdal1 and M. Rypdal1

1Department of Mathematics and Statistics, UiT The Arctic University of Norway, Norway

Correspondence to: Kristoffer Rypdal (kristoffer.rypdal@uit.no)

Science is comprised of the creative process of formulating new hypotheses and the systematic attempt to refute these hypotheses by testing them against observation. For the latter it is not sufficient to demonstrate that observations are con-

- sistent with the hypothesis according to some prescribed test, one also must make sure that the observations are inconsistent with other plausible (null-) hypotheses. In the present context this is particularly clear, because the statement that a response is nonlinear in itself is a negation. The main propo-
- sition in the paper by Lovejoy and Varotsos (L&V) is that the response is *not linear*. Thus, the only valid way of testing this statement against the data is to demonstrate that the linearity hypothesis is rejected by the data.

In section 2.4 (Fig. 2) and section 3.3 (Fig. 4) of our com-45 ment (R&R-C) we demonstrate that a linear response is consistent with the data. The reply of Lovejoy and Varotsos (L&V-R) only deals with section 3 of R&R-C, so we shall restrict ourselves here to the question of linearity and intermittencies. 50

The issue of multifractal (clustered) intermittency versus non-Gaussian Lévy processes and their long-memory derivatives was discussed at depth in a paper we recently published in Earth System Dynamics, with Shaun Lovejoy as a very active referee.1 We find it strange that L&V-R do not refer to 55 this paper and the associated discussion.

Our test presented in Fig. 4 is deliberately extremely simple. It can be appreciated by anyone, without understanding of the analysis method. We have just employed exactly the same analysis method as L&V (using Lovejoy's computer routines) on the data from a very simple *linear* response

puter routines) on the data from a very simple *linear* response model, a damped harmonic oscillator. The results of the analysis are indistinguishable from L&V's results from the same analysis of output from the ZC-model. This implies that the L&V results do not reject the linear response hypothesis.

*L&V-R* does not present any arguments against the validity of this test.

**Page 1, paragraph 2**: L&V-R cite two papers that are supposed to demonstrate the nonlinearity of responses to "spiky" forcing in some climate models, and state that their contribution has been to quantify this. There are many other papers that find very weak nonlinearities (see the paper by Andrews et al. and references therein). However, we do not claim that the response to such forcing impulses is linear - it seems quite plausible that they are not. Our claim is that the analysis by L&V does not reject such a claim, and by no means represents a "quantification of the nonlinearity."

They also write "The L&V method...simply exploits the known fact that linear transformations of a time series can only make linear changes to the exponent scaling function  $\xi(q)$ , they cannot affect the nonlinear part that is associated with the intermittency." In section 3.1-3.2 of our comment (R&R-C) we show that this is true only if the following three conditions hold: (I) The linear transformation can be represented by a power-law response function. (II) The structure functions are power-laws for all q. (III) Internal variability is negligible. Moreover, in section 3.4 we demonstrate by an example (the damped, harmonic oscillator) that the scaling function changes radically under this linear transformation when it is computed from the trace-moment analysis.

**Page 1, paragraph 3**: Here L&V-R defends the trace moment analysis. The method is supposed to be effective particularly because "... it removes the linear qH term in the structure function so that the linear K(q) part can be studied directly." We can't understand that subtraction of a straight line from a curved graph represents anything significant. The method is based on implicit assumptions that the underlying process *is* multifractal (that the structure functions are power-laws) with a distinct "outer scale" which defines the

<sup>1M. Rypdal and K. Rypdal, Late Quaternary temperature variability described as abrupt transitions on a 1/f noise background, Eart Syst. Dynam., 7, 281-293, 2016. doi: 10.5294/esd-7-281-2016. http://www.earth-syst-dynam.net/7/281/2016/esd-7-Discussion: http://www.earth-syst-dynam.net/7/281/2016/esd-7-281-2016-discussion.html

scale range where the power-law scaling holds. The powerlaw exponents (the slope of each trace moment of order qin a log-log plot) is determined by fitting straight lines in the log-log plot under the constraint that they all converge at

- the same time scale (the outer scale). In many cases (L&V Fig. 6) these lines are poor fits to the actual trace moments, 125 signifying that the time series are *not* multifractal. Our main problem with the trace moment method is that it is automatised to classify any non-Gaussian time series as multifractal.
- And by not devising error bars due to finite sample size, also 80 time series that are realisations of monofractal processes will 130 be classified as multifractals, since in single finite size realisations of the process there will always be deviations from the perfect scaling.
- In this paragraph L&V-R also write: "Had R&R noted this 85 fact (that removing the linear qH term allows K(q) to be 135 studied directly) they would not have bothered to develop the linear analysis (section 3.3) which is irrelevant to the trace moment analyses presented in L&V and to its conclusions."
- The conclusions of L&V are true only if conditions (I-III) 90 are valid, which we know the are not. How can this be irrel- 140 evant?

The phrase: "we would not have bothered to develop the linear analysis..." is disturbing. What does it signify? We

- are testing the linear response hypothesis by exploring its consequences and compare with data. Do L&V contend 145 that this is an incorrect or irrelevant approach? What is the alternative?
- Page 1, last paragraph, and page 2, paragraphs 1 and 2: 100 There are many formulations here that make no sense to us, 150 and therefore is hard to comment on. What is "a linear analysis"? Trace moment analysis is neither linear or non-linear. Linearity is a property of the response model which is subject
- to testing. And what do L&V mean by "dominant statistics"? We cannot relate to these phrases unless the authors are more 155 explicit about their meaning - if there is any.

What we can read out of these paragraphs, however, is that L&V define the class of multifractals to encompass essentially all stationary stochastic processes (monofractals constitute a subclass). The class of strictly multifractal processes (which excludes monofractals) then includes all stationary 160 non-Gaussian processes. We recommend the seminal paper by Mandelbrot, Fischer, and Calvet2 as a reference for the definition of multifractal stochastic processes.

115

The genealogies of multifractals (the multiplicative cascades) and of Lévy processes (non-Gaussian, uncorrelated noise) are fundamentally different, and so are the typical dynamical mechanisms. To treat them as one single class deprives us of impor-

120

A Multifractal Model of Asset Returns, Cowles Foundation Discussion Paper # 1164, September 15, 1997,

tant tools of understanding the underlying mechanisms governing a process.

L&V-R show in their Fig. 1a three multifractal constructions and one volcanic signal, suggesting that it is impossible by inspection to distinguish the latter from the others. Well, we immediately did. Multifractals and Lévy processes are both intermittent, in the sense that they are leptokurtic (heavy-tailed PDFs) on the short time scales and converge to Gaussian on the long time scales. But the multifractal intermittency is clustered, which the Lévy process is not. If you're training your eye, it is usually easy to distinguish a Lévy process from a multiplicative cascade. If you believe there is no essential difference, you probably will not see it. The important matter, however, is not what you can distinguish by your eye, but what you can disclose by analysis. By construction the structure functions of the multiplicative cascades are perfect power laws (straight lines in a log-log plot), while this will not be the case for the volcanic signal (see Fig. 3c in **R&R-C**). If a scaling function like K(q) is constructed from the indiscriminating trace moment analysis all signals will appear as multifractal.

If one generates realisations of multifractal and Lévy processes, there will always be realisations that look very similar. In order to see the difference, one will have to do statistics. If L&V will send us the routine they have used to generate the multifractals, with the parameters used to generate the plots in Fig. 1a, we shall do this statistics and demonstrate the difference.

L&V seem to believe that they escape from a logical problem by extending the multifractal class to encompass processes with non-power law structure functions, but by this they only create a new one. For this wide class of processes it is not true that "linear transformations of a time series can only make linear changes to the exponent scaling function  $\xi(q)$ ," no matter how much they insist on the opposite. This was proven theoretically in R&R-C section 3.3 and by the linear oscillator example in section 3.4. L&V's repeated disregard of these results does not make them less true.

If they maintain their assertion they have to show that our analysis in section 3.4 is wrong.3

Page 2, paragraphs 3 and 4, and L&V-R Fig. 1b: Fig. 1b is obvious. It follows from the non-Gaussianity of the signals. We agree that they are intermittent (bursty), and the volcanic signal is more intermittent than the response. But they are not multifractal. They could both arise from Lévy processes. The lower intermittency of the response signal are due to two different effects: (A) The memory in the response smears out the volcanic spikes. This is a linear effect. In Fig. 1e in this

165

<sup>2B. Mandelbrot, A. Fischer, and L. Calvet,

http://users.math.yale.edu/ bbm3/web\_pdfs/Cowles1164.pdf

<sup>3A Mathematica notebook which contains the computations leading to R&R-C Fig. 4 can be downloaded from https://dl.dropboxusercontent.com/u/12007133/Commentanalysis.nb

**K. Rypdal and M. Rypdal: Comment on Lovejoy and Varotsos**

- document we demonstrate that a transformation of the volcanic signal by a long-memory response kernel leads to a lower slope shown in L&V-R Fig. 1b. (B) Internal variability (noise) will also contribute to a flattening. In Fig. 1f we show the effect on the slope of adding a stochastic forcing in
- the response model, producing an internal variability. Hence L&V-R Fig. 1b can be explained perfectly from a linear response model.

We totally agree "that there is a huge difference in intermittency between R&R Fig. 4b and 4c." But that is our whole point!

> That huge difference in intermittency between forcing and response is produced by a linear response model, proving that L&V's assertion that "linear transformations of a time series can only make linear changes to the exponent scaling function" is wrong. It is wrong because conditions I-III described in R&R-C section 3 are not satisfied.

185

**R&R's "erroneous structure function analyses"**: This last part of L&V-C is yet another indication that L&V haven't read our comment properly. On lines 310-312 (Eq. (23)) we write explicitly that we compute the structure function from the cumulative summed forcing time series Y(t) = $\sum_{t'=0}^{t} F(t')$ , not from F(t) itself. In time series analysis this is the standard approach when structure functions are

195 computed from noise processes (H < 0). As L&V-R Fig. 2a shows, the structure function of the forcing time series F(t)itself is flat and reveal no other information than that the fluctuations are not growing with scale. Working on the cumulative sum (the "profile") is also the standard proce-200 dure in the DFA-analysis, which L&V-R employ in Fig. 2b.

- dure in the DFA-analysis, which L&V-R employ in Fig. 2b. The structure function defined in Eq. (23) can be written  $\hat{S}_q(\Delta t) = \Delta t^q \langle |\Delta F(t, \Delta t)|^q \rangle$ , where  $\Delta F(t, \Delta t)$  is the the moving average of F(t) over a window  $\Delta t$ . This is essentially the same as the Haar structure function of Lovejoy
- and Schertzer multiplied by  $\Delta t^q$ . Hence for a scaling noise for which  $\langle |\Delta F(\Delta t,t)|^q \rangle \sim \Delta t^{qH}$  the correspondence be-225 tween our structure function and the Haar structure function is  $\hat{S}_q(\Delta t) \sim \Delta t^q S_q^{\text{Haar}} \sim \Delta t^{(1+H)q}$ . What L&V-R plot in their Fig. 2b is  $(S_2^{\text{Haar}})^{1/2} \sim (\Delta t)^{-1} (\hat{S}_2(\Delta t))^{1/2}$ . From
- our Fig. 3c, which L&V-R claim is wrong, we find that  $\hat{S}_2(\Delta t) \sim \Delta t^1$  (the dashed line in the figure). From the equa-230 tion above this yields  $(S_2^{\text{Haar}})^{1/2} \sim \Delta t^{-1/2}$ , in good agreement with L&V Fig. 2b, and the spectrum shown in their Fig. 3 is of course also consistent with this.
- As mentioned above, working on the cumulative sum is standard in time series analysis, and was explicitly stated in R&R-C. Hence it does not give L&V much credit to overlook it. What is even more disturbing is that structure-function analysis of the signal itself and the cumulative sum was discussed in a response to Shaun Lovejoy in connection with
- another discussion paper in ESDD where Lovejoy is a ref-

(b)

Randomized volcanic forcing

(a)

Volcanic forcing

(b): A synthetic signal obtained from the volcanic signal in (a) by re-distributing the volcanic spikes randomly in time. (c) Tracemoment analysis of the volcanic signal used in the ZC-simulations. (d): Trace-moment analysis of the randomized volcanic signal in (b). (e): Shows the ration  $R = \langle \Delta F(\Delta t) \rangle / \sqrt{\langle \Delta F(\Delta t)^2 \rangle}$  for the volcanic signal used in the ZC-simulations (red) and for a linearresponse to this signal (purple). The linear-response model has a power-law Green's function determined by the exponent  $\beta = 0.8$ . (f): As in (e), but with a Gaussian white noise added to the forcing. The noise has a standard deviation  $\sigma = 0.3 \,\mathrm{Wm}^{-2}$ .

eree.4 Lovejoy complains that we do not cite thirty years old paper of his. What about reading and citing our contributions in ongoing public discussions?

As a final and explicit demonstration that trace moment analysis cannot determine whether the volcanic signal is a multifractal clustered time series or a non-Gaussian Lévy process we have made this analysis in Fig. 1a-d in this document. Fig. 1a shows the volcanic signal used in the ZCsimulations, and in Fig. 1b the same set of volcanic spikes with the time of the spikes chosen at random. Figs. 1c and 1d show that the corresponding trace moments are indistinguishable. This demonstrates that:

Automatised trace moment analysis only detects non-Gaussianity, not multifractal clustering.

<sup>4T. Nilsen, K. Rypdal, and H.-B. Fredriksen, Are there multiple scaling regimes in Holocene temperature records, Earth. Syst. Dynam. Discuss., 6, 1201, 2015, http://www.earth-syst-dynam-discuss.net/esd-2015-32/, see AC C610, 'response to Lovejoy'.

---

## Author Comment (AC2) · 9 Apr 2016

Manuscript prepared for Earth Syst. Dynam.
with version 2015/04/24 7.83 Copernicus papers of the LaTeX class coperni-
cus.cls.
Date: 9 April 2016

**Analysis of additivity in the NorESM model**

**K. Rypdal[1] and M. Rypdal[1]**

[1]Department of Mathematics and Statistics, UiT The Arctic University of Norway, Norway

*Correspondence to:* Kristoffer Rypdal  (kristoffer.rypdal@uit.no)

Lovejoy and Varotsos (L&V) analyse in their paper the ad-
ditivity of the response to solar and volcanic in the Zebiak-
Cane (ZC) model. There were at least three drawbacks with
these data. The ZC model is not representative for the global
temperature response, the data analyzed had been averaged
over 100 realisations, and there were no control runs avail-
able to assess the magnitude of internal variability. They also
analysed data from the NASA GISS-E2-R model, but here
they lacked the full suite of simulations with solar-only, vol-
canic only, and solar+volcanic forcing, and hence the could
not perform the test of the addititivity of responses on a full-
blown GCM.

We showed in our comment that L&V's analysis on the
additivity on the ZC-model is flawed. Now we have been
able to acquire a full suite of millennium-long simulations
for the NorESM Earth System Model, which is part of the
CMIP5 ensemble. More specifically, we have analysed solar-
only, volcanic only, solar+volcanic+anthropogenic, and con-
trol runs for the 900 yr period 935-1834 CE. We have omit-
ted the period after 1835 CE to minimize the anthropogenic
forcing in the full forcing simulation, and treat this as a so-
lar+volcanic simulation.

The global temperature responses and their corresponding
Haar fluctuation functions are given in Figure 1. It is remark-
able that all Haar fluctuation curves are almost flat, corre-
sponding to $H \approx 0$ or $\beta \approx 1$, i.e., to a so called $1/f$-noise.

In Figure 2 we have plotted in the same panel the Haar
fluctuations for the solar+volcanic (total) forcing (red), for
the summed responses to solar and volcanic forcing (blue),
and for the control run (magenta). Observe that the responses
to solar and volcanic forcing add up to the response of the
combined forcing. *The subadditivity claimed by L&V is com-
pletely absent*. We also observe that the internal variability
represented by the control run is quite strong. The standard
deviation of the internal variability is 2/3 of the variability of
the signal with solar+volcanic forcing. Moreover, the internal
fluctuations are almost equally strong on long-time scales as
on short time scales, contrary to what has been claimed by
L&V.

[Figure]

**Figure 1.** Left column: the response signals in the NorESM model.
Right column: the corresponding Haar fluctuation functions.

[Figure]

**Figure 2.** Red curve: Haar fluctuation of the response to solar + volcanic forcing. Blue curve: the Haar fluctuation of the summed solar and volcanic response. Magenta curve: Haar fluctuation of the control run.

**Acknowledgements.** This comment was supported by the the Norwegian Research Council (KLIMAFORSK programme) under grant no. 229754. We are grateful to Odd-Helge Otterå and the NorESM teams for providing the NorESM data.

---

## Short Comment (SC2) · 11 Apr 2016

[supplement omitted: unrelated document]

---

## Short Comment (SC3) · 13 Apr 2016

**On testing the additivity of Zebiak-Cane model response to volcanic and solar forcing.**

**Rebuttal of Section 2 of Rypdal and Rypdal 2016**

C. Varotsos[1], N. Sarlis[1], and S. Lovejoy[2]

[1]University of Athens, Dept. of Physics, University Campus Bldg. Phys. V, 15784 Athens, Greece
[2]Physics, McGill University, 3600 University St., Montreal, Quebec, Canada

**Introduction**

A rebuttal of Section 3 of Rypdal and Rypdal (2016) (R+R, below), in which we clarified some important points about intermittency in general and volcanic intermittency in particular, just appeared (Lovejoy and Varotsos, 2016a, as well as a further comment Lovejoy and Varotsos, 2016c). We now proceed to a rebuttal of Section 2 of R+R by examining the additivity of Zebiak-Cane (ZC) model response to volcanic and solar forcing using the data available from ftp://ftp.ncdc.noaa.gov/pub/data/paleo/climate_forcing/mann2005/mann2005.txt that have been also analyzed in the original paper by Lovejoy and Varotsos (2016b) (L+V below).

To situate the debate, recall that whereas at short enough time scales, when the external forcings are small enough, then theoretically we may expect the atmospheric response to be approximately linear, however, at long enough time scales, due to temperature – albedo feedbacks, the response is expected to become nonlinear. At the same time, it is possible that at long enough time scales, due to quite different detailed surface and atmospheric interactions, that solar and volcanic external forcings combine nonlinearly. Indeed this was one of the motivations for the Mann 2005 modelling study. In our paper L+V, we used both Mann's Zebiac-Cane model (100 realizations) as well as NASA GISS E2-R (full blown GCM) simulations to argue that for scales longer than about 50 years that there was evidence that solar and volcanic forcings combine subadditively. The fundamental difficulty - that we noted - was that neither had the full suite of simulations (i.e. control runs, solar only, volcanic only and combined solar and volcanic responses) needed in order to definitively answer the question. In the case of the Z-C model, the missing element was the control runs which meant that the internal variability was only indirectly estimated. R+R's criticism was based on the possibility that the internal variability – if large enough - might lead to a spurious subadditivity where in fact there was none. Although we believe that the original paper adequately answered this possibility, since this R+R's criticism, we have gone back to the ZC model and found a nearly 200 year period (at the beginning of the simulation series) where there was exactly zero volcanic forcing. Therefore this 200 year segment of the volcanic only response series was effectively a "control run" and could be used to estimate the magnitude of the internal variability. As we show below, at least for the Z-C model outputs, this settles the issue, vindicating the original L+V paper. However, in the last few days, R+R have found a totally different model and suite of results that they claim displays no subadditivity. Obviously – if valid - this new analysis of yet another model is an interesting contribution to the science since it would show that some models are subadditive while others are not. However, it is not directly relevant to the validity or otherwise of the original L+V paper.

**The R+R linear response null hypothesis test fails for the ZC model response**

R+R state (see lines 328-330): "*Our main focus in this comment, however, is not on the incorrect multifractal interpretation of the scaling analysis, but on the incorrect conclusions drawn from this analysis when it comes to nonlinearity in the response.*", here we deal with their main focus. Along these lines, we will check the validity of the linear response null hypothesis test that R+R suggested. In brief, by employing the notation "$\Delta$" for the fluctuation and the subscripts s, v, and s+v for solar, volcanic, and combined solar and volcanic response, respectively, R+R suggest the following test:

If $\Delta T_{s+v}-\Delta T_s-\Delta T_v\equiv\Delta\varepsilon$, i.e., their Eq.(6), then $\Delta\varepsilon$ has the same distribution as $\sqrt{3}\ \Delta E$ (cf. their Eq.(7)), where E is the internal variability which is unaffected by the forcing (see R+R lines 61-62). E is defined though their Eq.(1):

$$T(t)=T^{det}(t)+E(t), \qquad T^{det}(t)=\hat{L}[F(t)], \qquad \textbf{[1]}$$

where $\hat{L}[F(t)]$ is the linear response operator to the global forcing F(t). In their notation (c.f. they use a lower case epsilon ($\epsilon$) that is different from ε whereas we use capital E for the reader's convenience) one should have: " $\Delta\varepsilon \overset{d}{=} \sqrt{3}\,\Delta\epsilon.$ " (which is their Eq.(7)) for linear response and they conclude (see R+R lines 92 to 96) that:

"*Hence, a prediction based on the linear response hypothesis is that the difference between the temperature driven by combined solar and volcanic forcing and the sum of the temperatures driven by solar and volcanic forcing is realisation of a noise process which is $\sqrt{3}$ times the internal variability process. In the next subsection we shall test this prediction on the data from the ZC model. If the prediction is inconsistent with the data the linear response hypothesis is rejected for this model.*"

We will show below that the above test suggested by R+R fails.

According to their Eq.(9) (R+R line 126), for solar forcing we have:

$$E(t)=T_s(t)-\Delta T^{det}_s(t,\Delta t) \qquad \textbf{[2]}$$

where $\Delta T^{det}_s(t,\Delta t)$ is linearly related to the solar forcing $F_s(t)$ (see R+R Eq.(8) in line 117) through their Eq.(8):

$$\Delta T^{det}_s(t,\Delta t)=-S\ \Delta F_s(t-\tau,\Delta t) \qquad \textbf{[3]}$$

where S is the climate sensitivity and $\tau$ some time lag. Using now the fact (that was not noticed earlier!) that during the first 195 years of the ZC modelling the volcanic forcing is $F_v(t)=0$, Eqs.**[2]** and **[3]** applied for the volcanic response during the same 195 years yield:

$$\Delta E=\Delta T_v \qquad \textbf{[4]}$$

Hence, we can obtain $\Delta E$ from Eq. **[4]** and examine the validity of the linearity of the model by comparing whether $\Delta\varepsilon$ ($=\Delta T_{s+v}-\Delta T_s-\Delta T_v$) equals to $\sqrt{3}\Delta E=\sqrt{3}\Delta T_v$ for the first 195 years of the available ZC modelling.

Figure 1 shows that for the period of the first 195 years $\Delta\varepsilon$ (= $\Delta T_{s+v}$-$\Delta T_s$-$\Delta T_v$) is well below $\sqrt{3}\Delta E$ when using the Detrended Fluctuation Analysis (DFA) and hence the suggested test of linearity by RR fails. It also shows that even when considering all the 1000 years of the simulation $\Delta\varepsilon$ is well below $\sqrt{3}\Delta E$.

[Figure]

**Figure 1** Results of the DFA when using Eq.**[4]** to estimate the internal variability E. Both for the period of the first 195 years (blue) as well for the whole period of 1000 years (cyan) $\Delta T_{s+v}$-$\Delta T_s$-$\Delta T_v$ (=$\Delta\varepsilon$) is well below $\sqrt{3}\Delta E$ = $\sqrt{3}$ $\Delta T_v$ (thickest red line, top).

The same results are obtained when using the Haar wavelet analysis employed by L+V which are shown in Figure 2.

[Figure]

**Figure 2** Results from the Haar wavelet analysis used by L+V for the same time-series as in Fig.1, (in degrees C). We observe that for scales up to $10^{1.7} \approx 50$ years the RMS structure function of $T_{s+v}-T_s-T_v$ (either for the first 195 years or for the whole study period) never exceeds $\sqrt{3}\Delta E$ (the thickest red line) but rather remains close to $\sqrt{2}\Delta E$.

Hence, the linearity check suggested by R+R fails pointing to the non-linearity of the responses as suggested by L+V.

Other deficiencies of R+R are: a) they make use of statistical independence in their Eq. (12) - thus increasing the effect of the internal variability E to the ratio R - while they criticize L+V for doing so (see lines 162-164) and b) they claim (in lines 164-166) that L+V admit that their analysis is wrong, which of course is not the case.

Finally, in the references section of R+R the correct doi for L+V is 10.5194/esd-7-133-2016.

**References:**

Lovejoy, S., and C. Varotsos (2016a), Interactive comment on "Comment on "Scaling regimes and linear/nonlinear responses of last millennium climate to volcanic and solar forcing" by S. Lovejoy and C. Varotsos" by K. Rypdal and M. Rypdal, *Earth Syst. Dynam. Discuss.*, doi:10.5194/esd-2016-10-SC1.

Lovejoy, S., and C. Varotsos (2016b), Scaling regimes and linear/nonlinear responses of last millennium climate to volcanic and solar forcings, *Earth Syst. Dynam.*, 7, 1–18, doi:10.5194/esd-7-133-2016.

Lovejoy, S., and C. Varotsos (2016c), Trained Eye deceived by fractal clustering, Interactive comment on "Comment on: Scaling regimes and linear/nonlinear responses of last millennium climate to volcanic and solar forcing by S. Lovejoy and C. Varotsos" by K. Rypdal and M. Rypdal , http://www.earth-syst-dynam-discuss.net/esd-2016-10/esd-2016-10-SC2-supplement.pdf, Interactive comment on Earth Syst. Dynam. Discuss., Earth Syst. Dynam. Discuss. doi: doi:10.5194/esd-2016-10, 2016.

Rypdal, K., and M. Rypdal (2016), Comment on "Scaling regimes and linear/nonlinear responses of last millennium climate to volcanic and solar forcings" by S. Lovejoy and C. Varotsos, *Earth Syst. Dynam. Discuss.*, doi:10.5194/esd-2016-10, 2016.

---

## Author Comment (AC3) · 17 Apr 2016

Manuscript prepared for Earth Syst. Dynam.
with version 2015/04/24 7.83 Copernicus papers of the LaTeX class copernicus.cls.
Date: 17 April 2016

**Reply to the Lovejoy and Varotsos comment entitled "Trained eye deceived by fractal clustering"**

**K. Rypdal**[1] **and M. Rypdal**[1]

[1]Department of Mathematics and Statistics, UiT The Arctic University of Norway, Norway

*Correspondence to:* Kristoffer Rypdal  (kristoffer.rypdal@uit.no)

**Abstract.** This comment from L&V contains no substantiated arguments which invalidate anything in our comment article or in our first reply. It is a lengthy collection of unsubstantiated and erroneous claims which obscures the real issue, which is:

*Have L&V presented valid tests which prove that the temperature response in the climate models is inconsistent with a linear response model?*

L&V are in denial about the most obvious facts. One of these is that adding a noise to a intermittent signal will reduce the intermittency. The claim of theirs that is most relevant to the linearity issue is that the statistical uncertainties are so large that they overshadow the intermittency-reducing effect of curved structure-function plots and internal variability. This assertion is unsubstantiated, false, and bizarre. If statistical errors were this important they would invalidate all L&V's results from their original paper, and we would have used it against them. *Moreover, in our previous reply, a code was made available for L&V to check this for themselves, so they have no reason for making such a claim.*

In the discussion of how to define multifractality in stochastic processes L&V confuse the concept of a Lévy *process* with that of a Lévy *flight*, and disregard the definition coined by Mandelbrot, Calvet and Fisher (MCV) in 1997. MCV define a multifractal time series as one with power-law structure functions, which implies temporal dependence in the data. L&V associate multifractality with fractal properties of the *image set* $\{X(t)|t = 1, 2, \ldots\}$ of the time series, with the result that temporal dependence (clustering in time) will not be necessary property of a multifractal. For instance, with the L&V definition, random shuffling of a time series will not change the multifractal properties. The trace moments used by L&V to estimate multifractal intermittency *do* change with shuffling, hence trace moments measure something else than multifractality.

All these paradoxes are resolved by defining a multifractal time series as one with power-law structure functions. With this definition one has to accept that Lévy processes (non-Gaussian white noise) are not multifractals, and that moment-based estimators yield curved scaling functions whose curvature depends on a subjectively chosen scale range for fitting a straight line to curved trace moments (spurious multifractality).

L&V claim that we "misunderstand" multifractality and trace moment analysis. We respond by elaborating on our understanding of multifractality in an appendix. In the context of linearity testing, however, our understanding of the more arcane aspects is largely irrelevant. We know how the trace moment routine works and we can test the effect various properties in the data have on the intermittency estimates. Since trace moments is the only intermittency estimator applied by L&V our conclusions are valid even if we treat it as a "black box."

Below, we respond to L&V's reply in a chronological manner, to make sure that everything is addressed.

**1 Reply to "Summary"**

L&V: "...we quantified something - we thought - quite straightforward, the fact that the response of the atmosphere to volcanic forcing is nonlinear."

R&R: In our attached Figure 1a we show the Gao volcanic forcing and the global temperature responses in the NorESM model. It is not at all clear that strong volcanic forcing spikes give weaker responses than weak forcing spikes as claimed by L&V. The instantaneous responses seem quite proportional to the strength of the forcing.

L&V: "The basic fact that a linear transfer function (Green's function can only make a linear modification to the structure function exponent $\xi(q)$ has been known for some time and is even not contested by R&R."

R&R: Wrong! We *do* contest that assertion, because it is only valid under conditions I-III in our comment. L&V fail to state the conditions of validity. We proved in Sect. 2.4 in our comment that it is true only under these conditions which have been spelled out very clearly in our original comment and in our first reply, but L&V continue to ignore them. We repeat: the statement holds only if

(I): The Green's function is a power-law $G(t) \sim \Delta t^{\beta/2-1}$ on all interesting scales.
(II) The exponent $\xi(q)$ exist, i.e., if the structure functions actually *are* power laws on all interesting scales.
(III) The "response signal" does not contain a component from internal noise which will influence the intermittency estimates.

Condition II and III are the most important here. In our attached Figure 2 and 3 we demonstrate that deviations from power-law scaling in the structure function (condition II) makes the intermittency estimates depend on the particular scale range used for fitting a straight line to a curved graph. These figures are taken from a recent paper in ESD which was revived by Shaun Lovejoy. The breakdown of condition III is clearly illustrated in attached Figure 1. Figure 1b is a signal composed of two components. One is the volcanic forcing signal (red in Figure 1a) normalised such that the magnitude of the large volcanic spikes roughly match those of the volcanic response signal in NorESM (grey in Figure 1a). This signal can be thought of as the instantaneous response to the stochastic forcing. The other component is the internal variability represented by a control run. This composite signal represents a trivial linear transformation (multiplication by a normalization factor) plus a signal representative for the internal variability. Figure 1c,d shows the structure functions (SFs) and the scaling function for the Gao volcanic forcing computed from straight lines fitted to the SFs in the range displayed in Figure 1c. According to the assertion of L&V (who believe condition III is irrelevant), the intermittency shown by the curvature of the scaling function

in Figure 1d should be preserved in the scaling function for the composite signal shown in Figure 1g, but it is not. The latter signal is almost non-intermittent due to the "contamination" from the internal noise. This contamination explains the reduced intermittency observed in the the response to the volcano forcing shown in Figure 1e,d. This proves that:

> *nonlinearity in the response is not required to explain the difference in intermittency between forcing and response.*

L&V: The key limitation of the analysis was the existence of a single time series for each, and these were over finite ranges of time scales... These are the true limitations of our analysis and conclusions. R&R's hypotheses I-III are thus irrelevant as indicated in our response.

R&R: In Figure 1 we showed that condition III alone is sufficient to explain the entire difference in intermittency between forcing and observed response. The same was shown in our linear oscillator example in our comment article. In their efforts to escape from this conclusion, the authors now claim that the errors due to finite sample size are so large that they overshadow this intermittency-reducing effect of internal variability, and render this effect irrelevant. But if these errors were this large they necessarily would make the observed difference in intermittency statistically insignificant. A code has been made available for L&V to check that the statistical uncertainty is small. Hence,

> *the finite-sample error argument L&V is false and, if it were true, would invalidate their own conclusions.*

**2 Response to "The untrained eye works well"**

We stressed in our previous reply that the "trained eye" of course isn't a substitute for data analysis. But it is a fundamental principle in data analysis to perform a careful inspection of the data. For instance, the inspection of the signals of volcanic forcing and response put together in attached Figure 1a provides important information about the linearity in the instantaneous response to volcanic spikes that is not so easily quantified by analysis.

In the second paragraph on page 2, L&V suggest that we interpret "clustering" as synonymous with "fractal" (non-integer fractal dimension). This is a misunderstanding. We never expressed that idea. The discussion on pages 3 and 4, and Figs. 2-5, is based on two fundamental misconceptions:

1. L&V do not distinguish between a *Lévy process* and a *Lévy flight*. They write: "Recall that Levy processes have long power law tails on their probability distributions." This is incorrect. Lévy processes is a broad class of processes with independent increments. The

[Figure]

**Figure 1.** Analysis of global temperature responses in the NorESM model. (a): the Gao volcanic forcing (red) normalized such that the larges spikes are approximately equal to the spikes of the response signal (black). (b): the red curve in (a)+the control-run temperature signal. (c): structure functions (of cumulative sum) of volcano forcing. (d): scaling function derived from (c). (e): structure functions of the volcanic response. (f) scaling functions derived from (e). (g): structure functions of the signal in (b). (h): scaling function derived from (g). The red line arise from fitting straight lines in the entire scale range plotted, 4-128 yr. The blue line is from fitting only in the scale range 16-128 yr. It shows weak intermittency in both cases, but also that estimated intermittency depends on the scale range chosen for fitting. *The difference in curvature (reduction of intermittency) between (b) and (h) is exclusively caused by the addition of the internal noise represented by the control run.*

theory of Lévy processes was developed in the 1920s and 1930s, and we advice L&V to take a look at the excellent review of Appelbaum (2004). L&V are thinking of *Lévy flights*, which is just a small subclass of the Lévy processes where the PDF is so heavy-tailed that the variance is infinite. There is no reason to assume that any of the processes under discussion here are that heavy-tailed.

2. Their discussion revolves around clustering in the *image set* of the random variable; $\{X(t)\}|t=,1,2\ldots\}$ for a Lévy flight, while we are discussing the clustering of

spikes *in time* of spikes created by a *Lévy noise* (which is the increments of a Lévy process). The image set contains no information of the timing of the fluctuations, so shuffling of the data in time makes no change in the image set.

If the image set $\{X(t)\}|t=,1,2\ldots\}$ is what defines multifractality we face disturbing implications. One is that dependence in the data will be irrelevant. Such dependence (clustering of spikes in time) is the most prominent feature of a multiplicative cascade construction (e.g., a $\beta$-model). If we generate such a multifractal, and then perform a random shuffling of the data in time, we convert the data into realisations of a Lévy process. As we understand L&V, they consider the shuffled time series to be realisations of the same multifractal as obtained from the $\beta$-model. But we could also construct realisations of this Lévy process by drawing random numbers from the same non-Gaussian PDF (a Lévy-noise construction). Hence, if L&V are right the time series generated from the two very different constructions should possess the same multifractal intermittency. This is discussed in detail in the appendices of Rypdal and Rypdal (2016),[1] where Lovejoy was a very active referee. We find it strange that L&V do not refer to this paper and the associated discussion. There, we proved analytically that shuffling creates curved structure functions in a log-log plot, and in attached Figure 3 we demonstrate this by performing the constructions numerically. From this figure we observe that the structure functions and trace moments are different in two constructions which according to L&V represent the same multifractal. Hence, *we have to conclude that the trace moment analysis does not detect correctly the multifractality.*

The root of this paradox his the association of multifractality with the image set. The abstract geometric definition of multifractals is useful for theoretical purposes, but for estimation from data we have to resort to the moment-based approaches (the link between the two is the Legendre transform, but is subtle when we deal with finite data sets. We discuss this in some detail in in the appendix). This moment-based approach is applied by Mandelbrot, Fischer, and Calvet (1997)[2], where a multifractal is defined as one with power-law structure functions (a more correct term for the structure-function estimates is *empirical moments*[4]). This is a reasonable definition, because the scaling function and the intermittency parameters can only be estimated in a unique way when the structure function are straight lines in a log-log plot.

[1]M. Rypdal and K. Rypdal, Late Quaternary temperature variability described as abrupt transitions on a 1/f noise background, Earth Syst. Dynam., 7, 281-293, 2016, doi: 10.5294/esd-7-281-2016. http://www.earth-syst-dynam.net/7/281/2016/esd-7-281-2016.pdf

[2]B. Mandelbrot, A. Fischer, and L. Calvet, A Multifractal Model of Asset Returns, Cowles Foundation Discussion Paper # 1164, September 15, 1997, http://users.math.yale.edu/ bbm3/web_pdfs/Cowles1164.pdf

If they are curved these estimates will depend on the chosen fitting range. With this definition it can be proven that multifractal intermittency is associated with temporal dependence in the data.[1] Loosely speaking, multifractality implies that spikes are not randomly distributed in time, but are clustered in groups along the time axis.

Attached Figures 2 and 3b demonstrate that estimates from structure functions and trace moments yield considerable intermittency for non-Gaussian Lévy processes. According to the definition of Mandelbrot et al., these processes are not multifractals. This is the justification of our statement; *"trace moment analysis only detects non-Gaussianity, not multifractal clustering."* This answers L&V's "Minor comments, point 3."

[Figure]

**Figure 2.** (a): The increments of a jump-diffusion process shown in (b). This is a non-Gaussian independent noise process. (b): A realisation of a jump-diffusion process, and the cumulative sum of the signal in (a). This process is the sum of a Brownian motion and a Poisson jump process as described in Appendix B of Rypdal and Rypdal (2016).[1] The jump distribution is Gaussian with a standard deviation that is ten times greater than the standard deviation of the increments of the Brownian motion. (c): $S_q(\Delta t$ for $q = 1, 2, 3$ for the jump-diffusion process as computed from a large ensemble of realisations of the process. (d): Scaling function $\zeta(q)$ estimated from structure functions like those in (c). The red line is estimated by computing the slope of the structure-function curves on the longest time scale ($\Delta t = 500$). The blue curve is estimated from the slopes at the shortest time scale ($\Delta t = 1$). The black curve by estimating the slope of the straight line drawn between the end points of the structure-function curves. *This demonstrates that for processes with structure functions that are not power laws, and according to the MFC definition are not multifractals, the estimated intermittency depends on the scaling regime chosen for fitting.*

[Figure]

**Figure 3.** (a): realisation of a multiplicative cascade process. (b): the same process after random shuffling of the data. These two time series have the same image set, and according to L&V they have the same multifractal properties. (c) and (e): structure functions and trace moments of the realisation in (a). (d) and (f): the same for the realisation in (b). *Note the curvature of the plots in (d) and (f) for $q \neq 2$. The straight plot for $q = 2$ is a consequence of the independence in the time series in (b). The curvature of the plots influences the intermittency estimates and creates "spurious multifractality."*

Figure 5 in the reply of Lovejoy and Varotsos show that estimated scaling is similar for a Lévy process and a multifractal process. This is well known. See for instance the work of Neumann (2010)[3] or Heyde and Sly (2008).[4] This is called spurious multifractality; standard estimators may lead the scientist to conclude that a process has the characteristic of a multifractal, when in fact it does not.

> *Although the definition of multifractality is interesting to some, and important in some contexts in Earth system dynamics, it is largely irrelevant to the conclusions in the paper by L&V. All conclu-*

[3]Neumann S.: Apparent/spurious multifractality of data sampled from fractional Brownian/ Lévy motions, *Hydrol. Process.*, 24, 2056-2067, 2010.

[4]Heyde, C., and Sly, A.: A cautionary note on modeling with fractional Lévy flights, Earth. Physica A., 387, 5024-5032, 2014.

*sions on intermittency there are drawn from the trace moment analysis, and what's important is which properties of the signal this analysis measures, and not how we define multifractality.*

**3   Reply to "minor comments"**

**3.1   The statistical test argument**

The arguments and analogies L&V present here are so bizarre that we will not try to argue against them. We shall just state our point of view, which we believe is mainstream in the philosophy of science.[5]

We agree with L&V that "it is impossible in principle to prove linearity from data or from numerics." In fact, it is commonly accepted that *no falsifiable statement about nature can be proven or verified*, since there is always a chance that a new observation or a new test may prove it false. On the other hand, *only one single observation or test may be sufficient to falsify a well-posed hypothesis*. Hence, we cannot prove linearity, but we can falsify it, because from the linearity assumption we can make predictions that can be tested against observation. If we can falsify that the response is linear, i.e., if we can demonstrate that observation is inconsistent with a linear response, then we have proven that the response his nonlinear. We can do this because nonlinearity is a negation of a falsifiable statement. For the same reason it is impossible to falsify nonlinearity. The strongest conclusion we can draw is that the observations and tests at hand so far have failed to reject the linearity hypothesis. It should also be said, however, that if a broad range of tests fail to falsify linearity, then we will gain strong confidence in the hypothesis that the response is linear (in the sense that nonlinearity is so weak that it cannot be detected). This is the law of induction.

> *L&V may not be aware of it, but the two tests they have employed in their paper are exactly of this type. Our criticism is not of the logic of this test, but of the way it was done, and how the results were presented.*

**3.2   Reply to "Linear oscillators"**

In our comment we demonstrate that the trace moments of a particular linear response to volcanic + stochastic forcing (using the Green's function of a damped harmonic oscillator) is very similar to those of the Zebiak Cane model, demonstrating that a linear response plus internal variability can give a low-intermittency output from a high-intermittency input. What L&V insinuate is that we have selected a realisation that looks like ZC-output and in this way obtained trace moments that accidentally are similar to that of the ZC-output. Maybe this is L&V's way of doing science, but it is

[5]Bird, A., Philosphy of Science, Routledge, 1998.

not ours. The oscillator model output and the trace moment output are very similar in different realisations, and the ensemble mean of the trace moments will be smoother but have the same structure and give very similar intermittency parameters. In our previous reply we gave a URL where L&V can download a Mathematica notebook with the necessary routines to check this for themselves.

The point with our demonstration, however, was not to produce a response that imitates the ZC-model, but to demonstrate that a linear model with a reasonable level of internal variability can provide a strong reduction of the output intermittency, contrary to the claims of L&V. Another demonstration of this is shown in attached Figure 1b,d,f as discussed previously in this reply. Here the linear filter was simply multiplication by a normalization factor, and the reduction of intermittency was exclusively caused by the internal variability component (condition III).

The acrobatics L&V perform to escape from the conclusion that follows from these examples are similar to, and equally ungraceful, as their finite-sample error argument in "Summary." And it is again a boomerang on themselves. If the ensemble uncertainty of the trace moments is so large that the reduction of intermittency we observe is accidental it could be so also in the the actual ZC-data.

**3.3   Reply to "Misunderstanding the trace moment analysis"**

This point has been responded to in section 2 of this reply.

**3.4   Reply to "point 4"**

It is hard to take L&V's complaint about our notation seriously. In this cross-disciplinary field there is *absolutely no standard notation*. Much of he notation used in Lovejoy's writing is quite alien to us, and is closely connected to his own methodology. Often we have reservations against these methods, so why should we adopt his particular notation? It is part of our profession to accept other notations, as long as they are properly defined, as it was in our case.

Then to the particular points: In turbulence the fluctuations of the velocity field is growing with spatial scale. Translated to time series this corresponds to *motions* ($H > 0$). This is why structure functions in turbulence are computed directly from the increments of the velocity field itself. In geophysical time series a common situation is that the signal is a *noise* ($H < 0$). Structure functions computed directly from a noise provide no useful information. The standard procedure is therefore to create a motion by forming a cumulative sum, and to form the structure function on this signal. It is not particularly logical to change the notation $S_2$ because of this procedure.

The last sentence under this point reads: "In any case...there is no "curved scaling function" to be "incorrectly interpreted by L&V" (compare Fig. 2 and 3 of L&Vr)."

This statement demonstrates again that L&V still don't understand how structure functions work. In L&Vr Fig. 2 the first and second order structure functions are computed directly from the noisy signal. These structure functions are completely flat (constant) for *any* signal for which fluctuations do not grow with scale. They are completely insensitive to those properties that create the curvature in the the structure functions of the cumulative sum. In Fig. 3 L&V plot only the square root of the second-order Haar structure function. The $q = 2$ structure function is the only structure function that is not curved in an uncorrelated noise. This was a major issue in the discussion with Lovejoy in ESDD associated with the recently published paper,[1] where a rigorous proof was given. It is also demonstrated in attached Figure 2.

**4   Reply to "Conclusion"**

In the first pararagraph of the conclusion L&V reiterate their confusion concerning Lévy processes vs. Lévy flights. Non-Gaussian Lévy processes, that do not possess the extremely heavy tails of Lévy flights, do *not* exhibit characteristics that are indistinguishable from time series constructed from multiplicative cascades. This is clearly demonstrated in our Figure 3, where the structure functions and trace moments are changed from straight lines to curves in log-log plots. The problem is that mindless use of these moment-based estimators to construct scaling functions and estimate intermittency parameters often give similar results. The statistics and structure functions do *not* "show that quite different multifractal production mechanisms can lead to very similar statistics." The statistics is different, the structure functions/trace moments are different, it is just the intermittency estimated from these that sometimes give similar results. Such estimates from non-power law moments are not unique and have no real meaning.

The second paragraph seems to reiterate that L&V consider statistical errors (due to finite sample size and only one realisation available for analysis) are so large that they render our conclusions about the effect of curved structure functions and internal noise invalid (statistical insignificant). Here they admit that this argument may boomerang on their own results, but for some unspecified reason they conclude that "there are no compelling arguments to doubt our conclusions." The fact is that the statistical error limitation applies in principle to L&V's tests, because they are limited to finite sample size and one single realisation. But simple error estimates indicate that this is not a serious limitation for these data. Our test with the linear oscillator model, however, does *not* have this limitation. We can run as many realisations as we want, and we have of course checked that statistical errors are insignificant and do not invalidate our conclusions. If L&V still contend that we are cheaters they should run our Mathematica routine and prove it.

**Appendix A: Multifractals and Lévy processes**

**A1   A definition based on the $q$th moments**

Mandelbrot et al.[2] give the following definition of a multifractal stochastic process:

**Definition.** *Let $\stackrel{d}{=}$ denote equality in distribution. A stochastic process $X(t)$ with stationary increments is multifractal if*

$$X(t + a\Delta t) - X(t) \stackrel{d}{=} M(a)\Big(X(t + \Delta t) - X(t)\Big),$$

*where $M(a) \geq 0$ is a family of random variables satisfying the relation $M(ab) \stackrel{d}{=} M_1(a)M_2(b)$ with $M_1$ and $M_2$ being independent realizations of $M$.*

A self-similar process $X(t)$ is a special case of a multifractal process with $M(a) = a^h$ being a deterministic function of the scale $a$. For a multifractal process we have

$$\langle M(ab)^q \rangle = \langle M(a)^q \rangle \langle M(b)^q \rangle,$$

which implies that $\langle M(a)^q \rangle$ is a power law $\langle M(a)^q \rangle = a^{\zeta(q)}$. This implies the scaling relation

$$
\begin{aligned}
S_q(\Delta t) &= \langle |X(t + \Delta t) - X(t)|^q \rangle \\
&= \langle M(\Delta t)^q \rangle \langle |X(t + 1) - X(t)|^q \rangle \\
&= c_q \Delta t^{\zeta(q)},
\end{aligned}
$$

where $c_q = \langle |X(t + 1) - X(t)|^q \rangle$.

The functions $S_q(\Delta t)$ are called the structure functions, and most methods for estimating multifractality in observational data are based on structure functions, or on closely related constructions such as wavelet-based fluctuation functions. We stress that from the definition of Mandelbrot *et al.*, a multifractal process must have structure functions that are power-laws in the scale $\Delta t$, and only if this is satisfied is the scaling function $\zeta(q)$ defined.

Let us now recall the definition of a Lévy process.

**Definition.** *A stochastic process $X(t)$ is called a Lévy process if $X(0) = 0$ almost surly and*

1. *Increments are independent and stationary.*

2. *$X(t)$ is stochastically continuous with càdlàg paths.*

The second condition is a technical requirement that is not important for the considerations here, and one should think of a Lévy process as a continuous time random walk where the increments are not necessarily Gaussian. The simplest case of a Lévy process is the Wiener process (or Brownian motion) for which the increments $X(t + \Delta t) - X(t)$ *are* Gaussian. This process is self-similar with a self-similarity exponent $h = 1/2$, i.e. we have a linear scaling function $\zeta(q) = q/2$. Another class of self-similar Lévy processes

are the so-called Lévy flights for which the increments have $\alpha$-stable (heavy-tailed) distributions. For Lévy flights with stability parameter $\alpha < 2$ we have monofractal scaling $\zeta(q) = q/\alpha$ for $q < \alpha$, and the scaling function is not defined for $q > \alpha$ due to the heavy tails. However, the self-similarity relation $X(at) = a^h X(t)$ is valid with $h = 1/\alpha$. It is well known that if one attempts to estimate the scaling function for a Lévy flight using structure functions, then spurious multifractality is observed (Heyde and Sly, 2008). The extreme tails in the increment distribution of a Lévy flight are often unrealistic models of real world observations, and it is usually sufficient to consider non-Gaussian distributions for which all the moments exist. One can for instance use truncated Lévy flights[6], or any other non-Gaussian Lévy process.

**Proposition.** *A Lévy process with finite moments is not a multifractal stochastic process in the sense of Mandelbrot et al., unless it is the Wiener process.*

The proof is simply to show that Lévy processes that are non-Gaussian have structure function that are not power laws. This is done in the appendices of Rypdal and Rypdal (2016)[1]. For the purpose of modelling volcano forcing as a stochastic process (for instance as the increments of a multifractal process or as the increments of a Lévy process) we can draw the following conclusion:

*If the statistical properties of the volcano forcing signal are invariant under shuffling, then it is not consistent with a multifractal stochastic process in the sense of Mandelbrot et al.*

**A2   A definition based on the singularity spectrum**

An alternative way of defining what is meant by a multifractal stochastic process is via the so-called singularity spectrum. For a realization of the stochastic process $X(t)$ one can define the local Hölder exponents as

$$\gamma(t) = \liminf_{\substack{s \to t \\ s \neq t}} \frac{\log |X(s) - X(t)|}{\log |s - t|},$$

with the convention that $\log 0 = -\infty$. The definition ensures that $\gamma(t)$ exists for any realization of $X(t)$ for all $t$, and if we have asymptotic scaling

$$|X(t + \Delta t) - X(t)| \sim |\Delta t|^{\gamma'} \text{ as } \Delta t \to 0,$$

then $\gamma(t) = \gamma'$. For some stochastic processes there is essentially only one Hölder exponent $\gamma$. For instance, a realisation of a Wiener process has with probability one $\gamma(t) = 1/2$ for all time instances $t$.
* * *
[6]  Terdik, G., Woyczynski, W. A., and Piryatinska, A.: Fractional- and integer-ordered moments, and multiscaling of smoothly truncated Lévy flights. Phys. Lett. A, 348, 94-109, 2006.

The singularity spectrum for a realisation of a stochastic process is a function $f(\gamma)$ that specifies the fractal dimensions of the level sets of the function $\gamma(t)$:

$$f(\gamma) = \dim_H K_\gamma,$$

where $K_\gamma = \{t \in \mathbb{R} : \gamma(t) = \gamma\}$. Here $\dim_H(\cdot)$ denotes the Hausdorff dimension, and the convention is that the dimension of the empty set $\emptyset$ is $-\infty$. For a Wiener process we have $K_{1/2} = \mathbb{R}$ and $K_\gamma = \emptyset$ for $\gamma \neq 1/2$, and hence the singularity spectrum is

$$f(\gamma) = \begin{cases} 1 & \gamma = 1/2 \\ -\infty & \text{else} \end{cases}.$$

Since $f(\gamma)$ has a positive value for only one particular Hölder exponent, the Wiener process is often termed monofractal.

If a process is a multifractal in the sense of Mandelbrot *et al.*, so that its scaling function $\zeta(q)$ is defined, then one can in some cases express the relationship between the scaling function and the singularity spectrum via the Legendre transform:

$$f(\gamma) = \inf_q \{q\gamma - \zeta(q) + 1\}. \tag{A1}$$

If this relation holds, then a monofractal singularity spectrum corresponds to a linear scaling function, and hence to a self-similar process. And if the scaling function $\zeta(q)$ is strictly concave, then this corresponds to a non-trivial singularity spectrum where there is a range of $\gamma$-values for which the level sets $K_\gamma$ have positive Hausdorff dimensions. In other cases, such as for $\alpha$-stable Lévy processes (so-called Lévy flights), the relation in Eq. (A1) does not hold since $\zeta(q)$ is not defined for $q > \alpha$. However, it has been shown by Heyde and Sly (2008) that the estimated (using standard methods) scaling function of a Lévy flight is

$$\hat{\zeta}(q) = \begin{cases} q/\alpha & q < \alpha \\ 1 & q > \alpha \end{cases}, \tag{A2}$$

and it has been shown by Jaffard (1999)[7] that the singularity spectrum has the form

$$f(\gamma) = \begin{cases} \alpha\gamma & q \in [0, 1/\alpha] \\ -\infty & \text{else} \end{cases},$$

which is the Legendre transform of the expression in Eq. (A2). In this sense, one might say that the multifractal formalism works for $\alpha$-stable Lévy processes. On the other hand, there are heavy-tailed Lévy processes that have monofractal singularity spectra, even though the estimated scaling functions are concave. Hence, one should be careful to interpret a signal as multifractal based on the estimated scaling function alone. In the next subsection we will list some results that show that a reasonable Lévy-process description of the volcanic forcing signal must have a monofractal singularity spectrum.
* * *
[7]Jaffard, S.: The multifractal nature of Lévy processes. Probability Theory and Related Fields ,114, 207-227, 1999.

**A3   The singularity spectra of Lévy processes**

A Lévy process $X(t)$ is conveniently characterized by the characteristic functions of the variables $X(t)$. These are defined as $\phi_{X(t)}(u) = \langle e^{iuX(t)} \rangle$. A consequence of the independence of increments is that the characteristic functions are of the form $\phi_{X(t)}(u) = e^{t\psi(u)}$, where the function $\psi(u)$ is called the Lévy exponent. In order for Lévy processes to be well defined in continuous time the variables $X(t)$ need to to be infinitely divisible, and a consequence of this condition is that the Lévy exponent is on the form

$$\psi(u) = \mu u i - \frac{\sigma^2 u^2}{2} + \int\limits_{|x|\geq 1} (e^{iux} - 1)d\nu(x)$$
$$+ \int\limits_{|x|<1} (e^{iux} - 1 - iux)d\nu(x),$$

where $\nu$ is a measure on $\mathbb{R} \setminus \{0\}$ satisfying the condition

$$\int \min\{1, x^2\}d\nu(x) < \infty. \tag{A3}$$

The measure $\nu$ is called the Lévy measure, and if it is zero, then $X(t)$ is a Brownian motion with scale parameter $\sigma$ and drift $\mu t$. If we assume that $\langle |X(1)| \rangle < \infty$, then

$$\int\limits_{|x|<1} x d\nu(x) < \infty$$

so that one can write

$$\psi(u) = (\mu - \mu')ui - \frac{\sigma^2 u^2}{2} + \int\limits_{\mathbb{R}\setminus\{0\}} (e^{iux} - 1)d\nu(x),$$

where $\mu' = \int_{|x|<1} x d\nu(x)$. Moreover, if the measure $\nu$ is finite, then one can define a probability measure $dP_J(x) = \lambda^{-1} d\nu(x)$, where

$$\lambda = \int\limits_{\mathbb{R}\setminus\{0\}} d\nu(x).$$

This gives

$$\psi(u) = (\mu - \mu')ui - \frac{\sigma^2 u^2}{2} + \lambda \int (e^{iux} - 1)dP_J(x),$$

and the corresponding process is called a jump-diffusion. If we have no Brownian component and no drift, i.e., if

$$\psi(u) = \lambda \int (e^{iux} - 1)dP_J(x),$$

then the $X(t)$ is called a Poisson jump process. The number $\lambda$ is the jump rate, and $P_J(x)$ is the probability density function for the jump-size.

The increments of a Poisson jump process is a better model for the volcano forcing signal than the increments of a Lévy flight (for which $\lambda$ is infinite). The reason for this is that the existence of a finite rate $\lambda$ is equivalent to the assumption that there are only a finite number of volcanic eruptions in any finite time interval. The probability density $P_J(x)$ for the jumps may well be heavy-tailed, and this does not affect the singularity spectrum as long as the rate $\lambda$ is finite. In fact, any such process has the following monofractal singularity spectrum (Jaffard, 1999):

$$f(\gamma) = \begin{cases} 0 & \gamma = 0 \\ -\infty & \text{else} \end{cases}$$

The results summarized in this subsection leads us to the following conclusion.

> If the volcanic forcing signal is modeled as the increments of a Lévy process $X(t)$, then since there are only a finite number of volcanic eruptions in any finite time interval, the process $X(t)$ has a monofractal singularity spectrum.

The discussion of wether a Lévy flight should be considered as multifractal or monofractal is hence irrelevant for the purpose of describing volcanic forcing.

**Remark.** Since most estimators of multifractality are based on structure functions (or similar constructions based on $q$th moments) it is our opinion that the definition of Mandelbrot *et al.* is the most useful. However we have to respect that other authors may use the term in different ways, and this should not be problematic as long as it is clear what is meant. In this discussion, however, we are not sure what Lovejoy and Varotsos mean by a multifractal stochastic process, and we therefore ask them to be precise on this point.

---

## Author Comment (AC4) · 17 Apr 2016

Manuscript prepared for Earth Syst. Dynam.
with version 2015/04/24 7.83 Copernicus papers of the LaTeX class copernicus.cls.
Date: 17 April 2016

**Reply to Lovejoy, Sarlis and Varotsos: "On testing the additivity of Zebiak-Cane model response to volcanic and solar forcing"**

**K. Rypdal**[1] **and M. Rypdal**[1]

[1]Department of Mathematics and Statistics, UiT The Arctic University of Norway, Norway

*Correspondence to:* Kristoffer Rypdal (kristoffer.rypdal@uit.no)

**Abstract.** Lovejoy, Sarlis, and Varotsos (LSV) replicate a test in our comment by using a different time series representing internal variability, and conclude that this test rejects the linear response hypthesis in the Zebiak-Cane model. This time series is the first 195 yr of the volcanic response record, during which there was no volcanic forcing. We demonstrate that the short length of this record creates large finite size uncertainties which render their result statistically insignificant. We also comment on some passages in their reply about physical paradigms, and on faulty statistical reasoning and apparent self-contradictions in L&V's writings.

**1 Reply to "Introduction"**

As a motivation for conduction these tests in the ZC model LSV write:

> "*To situate the debate, recall that whereas at short enough time scales, when external forcings are small enough, then theoretically we may expect the atmospheric response to be approximately linear, however, at long enough time scales, due to temperature - albedo feedbacks, the response is expected to become nonlinear. At the same time, it is possible that at long enough time scales, due to quite different surface and atmospheric interactions, that solar and volcanic external forcings combine nonlinearly.*"

A qualitative mental picture, a paradigm, of the physical mechanisms that govern the macro dynamics is of course an important guideline, but it can also become a straitjacket that restricts the range of alternatives that one is willing to investigate. Our favorite picture is quite different L&V's. We se no reason why responses should be more linear on short than on long time scales, in particular not the response to volcanic forcing, which is strong on short time time scales. LSV write about "atmospheric response" which is a vague concept that is impossible to quantify without specifying the physical variable and on which spatial and temporal scale it is measured. The response of local climatic variables on synoptic and seasonal scales to strong volcanic eruptions is certainly nonlinear, since it is dominated by hydrodynamical vortex structures. But on long time scales, the global temperature change will change in proportion to the change in heat content in the upper ocean, which again will change in proportion to the net radiative flux. The response in presence of feedbacks that modify the radiative flux, like albedo feedbacks, are not "expected to become nonlinear" by most climate modelers. They are typically modeled linearly. Consider, for instance, the linear energy balance model $d_t T = S^{-1}T + F$, and assume that the albedo decreases proportionally to $T$, such that the effective radiative flux can be written $F = F_0 - \alpha T$. The resulting equation is $d_t T = S'^{-1}T + F_0$, where the feedback has enhanced the climate sensitivity to $S' = (1 - \alpha S)^{-1}S$.

The ENSO phenomenon is probably a nonlinear mode in the climate system, and is part of the internal variability, even though it can be influenced by external forcing. The nonlinear nature of the oscillation makes it likely that the timing of El Niño events can be influenced by external forcing such as strong volcanic eruptions. But we find it less likely that the response on centennial time scales is nonlinear. This is the issue discussed in section 2 of our comment article. But whatever our prejudices are, proper tests is what should settle the issue.

**2   Reply to "The R+R linear response null hypothesis test fails for the ZC model response"**

In section 2 of our comment we presented two tests of linearity of the response in the ZC-model. In Sect. 2.3 we presented an alternative test of response additivity which involved an estimate of internal variability and its effect on the test. This is the only test that is addressed in LSV's reply. In Sect. 2.4 we replicated the test made in the original paper by L&V (which does not take internal variability into account). This replication failed to detect the subadditivity claimed by L&V and presented in a very confusing manner in their paper. It would be interesting to know weather L&V will dispute the correctness of Fig. 2 in our comment article, for instance by replicating that figure.

In our test in Sect. 2.3 we estimated internal variability from the solar forcing and response, using the entire 1000 yr time series. It involved fitting a 25 yr shifted linear response to the observed response signal and interpreting the residue as the internal variability. A weakness of this procedure is that it involves some assumptions about the slow linear response, but an advantage is that finite sample size uncertainty could be kept low because we used the entire time series. The mentioned assumptions reduces the strength of the test, so the only conclusion we could draw is that this relatively weak test could not reject the linearity hypothesis. We also wrote in the comment, that long control runs of the model could give us a better estimate of the internal noise and a stronger test.

In their reply, LSV suggest to use a different estimate of the internal noise, namely the first 195 yr of the volcanic-driven response time series. This is justified, since there was no volcanic forcing in this period. The drawback, however, is that an estimate of the Haar fluctuation from such a short time series is associated with much higher estimation uncertainty (finite-sample size errors). LSV make no attempt to demonstrate that the estimates of the difference $|\sqrt{3}\epsilon(t) - \varepsilon(t)|$ is significantly different from zero in a statistical sense. Such a test is easy to make by creating a Monte Carlo ensemble of time series containing 195 data points with statistical properties similar to those of the observed volcano response. The statistical scatter of the Haar fluctuations within this ensemble will give us information about the finite sample uncertainty of the Haar estimate. This is done in attached Figure 1, where the specifications of the Monte Carlo are desccribed in the caption. The figure shows that the difference between the Haar fluctuations of $\sqrt{3}\epsilon(t)$ and $\varepsilon(t)$ is smaller than this uncertainty in the interesting scale range $\Delta t > 10$ yr.

*This means that the deviation from linearity observed by LSV is statistically insignificant, and hence does not reject the linear response hypothesis.*

[Figure]

**Figure 1.** Brown bullets: Haar fluctuation function of $\sqrt{3}\epsilon(t)$, where $\epsilon(t)$ is the first 195 yr of the volcanic forcing record. Red bullets is Haar-fluctuation of $\varepsilon(t)$ as defined in the comment article of R&R. These two curves look similar to the corresponding curves in Fig. 2 in LSV. The crucial issue is whether the difference between these two curves is statistically significant. The green curves is a 500 member ensemble of fractional Gaussian noises (fGn's) with $H = -0.01$ ($\beta = 2H + 1 = 0.98$) and Haar fluctuation equal to that of $\sqrt{3}\epsilon(t)$ on 100 yr time scale. On time scale less than 10 yr the fGn is not a good model for the internal noise because of the ENSO dynamics, but on longer time scales the flat Haar-fluctuation curve suggests that an fGn with $\beta \approx 1$ is a crude statistical model of the internal variability. The scatter of the Haar fluctuation in this ensemble gives an idea about the statistical uncertainty of this volcano estimate of internal variability. This uncertainty is considerably larger than the estimate of $|\sqrt{3}\epsilon(t) - \varepsilon(t)|$ (the difference between the brown and the red curves), hence this difference is not statistically significant.

**3   Elementary errors of statistical reasoning and persistent self-contradicion in L&V's writings**

The last paragraph in the LSV reply reads:

"Other deficiencies of R+R are:
a) they make use of statistical independence in their Eq. (12) -thus increasing the effect of the internal variability E to the ratio R - while they criticize L+V for doing so (see lines 162-164) and
b) they claim (in lines 164-166) that L+V admit that their analysis is wrong, which of course is not the case."

Eq. (12) in our comment reads

$$R = \sqrt{1 + \frac{\langle|\Delta\varepsilon|^2\rangle}{\langle|\Delta T_{s+v}|\rangle^2}}.$$

No assumption of statistical independence was used up to that point. Note that $R = R(\Delta t)$, i.e., it is a generalisation of the $R(\Delta t)$-curve plotted in Figure 3 in the L&V paper, including internal noise. If we had set out to estimate this curve

we should not estimate $\langle|\Delta\varepsilon|^2\rangle$ by using $\sqrt{3}\epsilon(t)$ as an estimator, but rather the exact definition of $\Delta\varepsilon$ given by Eq. (6); $\Delta\varepsilon = \Delta T_{v+s} - \Delta T_v - \Delta T_s$. We did not plot such a curve, however, because our ambition was only to demonstrate that internal variability increase $R$, and to indicate the order of magnitude of that increase.

What L&V did in their paper was different. They used an approximation analogous to, but different from, ours to estimate Haar fluctuation curves, and this turned out to give a systematic bias in favour of subadditivity. Our approximation was concerned a noise process (internal variability), and the error is a finite size error, not a bias. The L&V approximation treated the volcanic and solar forcing as independent stochastic processes, while the appropriate way of dealing with these 1000 yr historical records (in particular the solar) is to treat them as deterministic signals. In that case omission of cross-terms have little justification.

After being pushed on this point by us in the ESDD discussion of their paper, L&V kept their original approximation in the text and in the caption of Figure 3, but included a revision of the $R$-curve in the figure, so that the figure became a hybrid that contains results both based on the invalid approximation and results that are not. They acknowledged that the approximation makes a difference, and in the text they added the following paragraph:

> "The reason for the difference is that the cancellation of the cross terms assumed by statistical independence is only approximately valid on single realizations, especially at the lower frequencies where the statistics are worse (even on a single realization, at any given scale - except the very longest - there are several fluctuations, so that there is still some averaging).

We interpreted this as an admission that the approximation is wrong. However, in (b) above they now state that this is not the case. They believe both results are correct, and thereby transcend the trivial realm of logic…

---

## Short Comment (SC4) · 26 Apr 2016

**Final comment on low frequency linearity, nonlinearity**

**C. Varotsos[1], S. Lovejoy[2], and N. Sarlis[1]**

[1]University of Athens, Dept. of Physics, University Campus Bldg. Phys. V, 15784 Athens, Greece
[2]Physics, McGill University, 3600 University St., Montreal, Quebec, Canada

Any readers who have followed the exchange of our opinion (http://www.earth-syst-dynam-discuss.net/esd-2016-10/esd-2016-10-SC1-supplement.pdf , http://www.earth-syst-dynam-discuss.net/esd-2016-10/esd-2016-10-SC2-supplement.pdf, http://www.earth-syst-dynam-discuss.net/esd-2016-10/esd-2016-10-SC3-supplement.pdf ) with those of Rypdal and Rypdal (2016a-e) (R+R, below) up to this point will have noticed that the linearity check, which was the main focus of R+R (2016a), fails since we showed (in SC3) that for scales up to $10^{1.7} \approx 50$ years the RMS structure function of $T_{s+v}$-$T_s$-$T_v$ (either for the first 195 years or for the whole study period) never exceeds $\sqrt{3}\Delta E$ (as suggested by R+R, 2016a) but rather remains close to $\sqrt{2}\Delta E$. Notably, R+R (2016e) did not explain this failure.

We take their AC3 response (R+R, 2016c), - the analysis of a completely different set of model runs (NorESM) – to be an indirect admission that our original paper was correct. If one day this new analysis of new model outputs is published in the peer reviewed literature, it could be a useful contribution to our understanding of approximate linearity in nonlinear climate models, but would not in any case invalidate the published analyses of the different model outputs in Lovejoy and Varotsos (2016).

**References:**

Lovejoy, S., and C. Varotsos (2016), Scaling regimes and linear/nonlinear responses of last millennium climate to volcanic and solar forcings, Earth Syst. Dynam., 7, 1–18, doi:10.5194/esd-7-133-2016.

Rypdal, K., and M. Rypdal (2016a), Comment on "Scaling regimes and linear/nonlinear responses of last millennium climate to volcanic and solar forcings" by S. Lovejoy and C. Varotsos, *Earth Syst. Dynam. Discuss.,* doi:10.5194/esd-2016-10, 2016.

Rypdal, K., and M. Rypdal (2016b), Interactive comment on "Comment on "Scaling regimes and linear/nonlinear responses of last millennium climate to volcanic and solar forcing" by S. Lovejoy and C. Varotsos" by K. Rypdal and M. Rypdal, Earth Syst. Dynam. Discuss., doi: 10.5194/esd-2016-10-AC1, 2016.

Rypdal, K., and M. Rypdal (2016c), Interactive comment on "Comment on "Scaling regimes and linear/nonlinear responses of last millennium climate to volcanic and solar forcing" by S. Lovejoy and C. Varotsos" by K. Rypdal and M. Rypdal, Earth Syst. Dynam. Discuss., doi:10.5194/esd-2016-10-AC2, 2016.

Rypdal, K., and M. Rypdal (2016d), Interactive comment on "Comment on "Scaling regimes and linear/nonlinear responses of last millennium climate to volcanic and

solar forcing" by S. Lovejoy and C. Varotsos" by K. Rypdal and M. Rypdal, Earth Syst. Dynam. Discuss., doi:10.5194/esd-2016-10-AC3, 2016.

Rypdal, K., and M. Rypdal (2016e), Interactive comment on "Comment on "Scaling regimes and linear/nonlinear responses of last millennium climate to volcanic and solar forcing" by S. Lovejoy and C. Varotsos" by K. Rypdal and M. Rypdal, Earth Syst. Dynam. Discuss., doi:10.5194/esd-2016-10-AC4, 2016.

---

## Short Comment (SC5) · 26 Apr 2016

**Summary of Lovejoy and Varotsos rebuttal (SC1, SC2, and SC3) to Rypdal and Rypdal comment and replies (AC1, AC2, AC3, and AC4)**

**S. Lovejoy[1], C. Varotsos[2], and N. Sarlis[2]**

[1]Physics, McGill University, 3600 University St., Montreal, Quebec, Canada

[2]University of Athens, Dept. of Physics, University Campus Bldg. Phys. V, 15784 Athens, Greece

Any readers who have followed the exchange of our opinion (http://www.earth-syst-dynam-discuss.net/esd-2016-10/esd-2016-10-SC1-supplement.pdf , http://www.earth-syst-dynam-discuss.net/esd-2016-10/esd-2016-10-SC2-supplement.pdf, http://www.earth-syst-dynam-discuss.net/esd-2016-10/esd-2016-10-SC3-supplement.pdf ) with those of Rypdal and Rypdal (2016a-e) (R+R, below) up to this point will have noticed that the discussion is now far from our original paper (Lovejoy and Varotsos, 2016) and that Rypdal and Rypdal (2016a-e) are unable to discuss the issues in simple and widely accessible terms. They constantly impute to us either irrelevant or unnecessarily - and inappropriately - precise mathematical assumptions and then try to demonstrate that these assumptions - that we do not make - are problematic. One gets the impression that they have difficulty distinguishing mathematics from geophysics.

The debate about linear versus nonlinear behavior clearly illustrates the difference between our two approaches. Our physically based approach asks under what conditions - over what range of fluctuation time scales and amplitudes - is a linear model *a useful, reasonable* approximation to the real world system. In comparison, R+R's attempt to solve the problem mathematically is futile if only because linearity is a special case of nonlinearity: it is therefore *in principle* impossible to numerically or empirically prove that a given system is linear. At most one can place upper bounds on the *degree* of nonlinearity, and even this requires further assumptions in order to restrict the possible types of nonlinearity to a manageable framework. In essence, R+R's approach represents a Platonic attempt to settle scientific questions by mathematics. In their latest response, their obsession with mathematics leads largely to a war of words with R+R constantly using needlessly restrictive definitions - notably of Levy processes and multifractals - and then accusing us of not using their definitions!

Up until now, we have endeavored to maintain the discussion at a level that will be of interest to the general ESD reader, but now we have reached the limits of what is possible in the confines of ESDD discussions: the exchange has become unproductive and it serves no purpose to continue.

**References:**

Lovejoy, S., and C. Varotsos (2016), Scaling regimes and linear/nonlinear responses of last millennium climate to volcanic and solar forcings, Earth Syst. Dynam., 7, 1–18, doi:10.5194/esd-7-133-2016.

Rypdal, K., and M. Rypdal (2016a), Comment on "Scaling regimes and linear/nonlinear responses of last millennium climate to volcanic and solar forcings" by S. Lovejoy and C. Varotsos, *Earth Syst. Dynam. Discuss.,* doi:10.5194/esd-2016-10, 2016.

Rypdal, K., and M. Rypdal (2016b), Interactive comment on "Comment on "Scaling regimes and linear/nonlinear responses of last millennium climate to volcanic and solar forcing" by S. Lovejoy and C. Varotsos" by K. Rypdal and M. Rypdal, Earth Syst. Dynam. Discuss., doi: 10.5194/esd-2016-10-AC1, 2016.

Rypdal, K., and M. Rypdal (2016c), Interactive comment on "Comment on "Scaling regimes and linear/nonlinear responses of last millennium climate to volcanic and solar forcing" by S. Lovejoy and C. Varotsos" by K. Rypdal and M. Rypdal, Earth Syst. Dynam. Discuss., doi:10.5194/esd-2016-10-AC2, 2016.

Rypdal, K., and M. Rypdal (2016d), Interactive comment on "Comment on "Scaling regimes and linear/nonlinear responses of last millennium climate to volcanic and solar forcing" by S. Lovejoy and C. Varotsos" by K. Rypdal and M. Rypdal, Earth Syst. Dynam. Discuss., doi:10.5194/esd-2016-10-AC3, 2016.

Rypdal, K., and M. Rypdal (2016e), Interactive comment on "Comment on "Scaling regimes and linear/nonlinear responses of last millennium climate to volcanic and solar forcing" by S. Lovejoy and C. Varotsos" by K. Rypdal and M. Rypdal, Earth Syst. Dynam. Discuss., doi:10.5194/esd-2016-10-AC4, 2016.

---

## Referee Comment (RC1) · Anonymous Referee #1 · 3 May 2016

In his role as referee of "Scaling regimes and linear/nonlinear responses of last millennium climate to volcanic and solar forcings", K. Rypdal already made many of the points that were then reiterated with M. Rypdal in this publication (see http://www.earth-syst-dynam-discuss.net/6/1815/2015/esdd-6-1815-2015-RR1.pdf). This new publication adds some detailed mathematics and some numerical analysis, but does little to illuminate the fundamental scientific debate about linearity/nonlinearity. Indeed, the entire debate that started with the original review of the Lovejoy Varotsos ESDD paper is not materially advanced by this new contribution.

[Figure]

Let us recall the interactions between L+V and K. Rypdal:

Oct. 23 2015, comments on the first draft of L+V

Dec. 15, 2015: L+V response.

Jan. 12, 2016, L+V revised ms submitted.

Jan. 21. 2016, K. Rypdal comments.

March 18, 2016: submission of the current paper "Comment on "Scaling regimes and linear/nonlinear responses of last millennium climate to volcanic and solar forcing" by S. Lovejoy and C. Varotsos"

Following this are no less than five exchanges:

SC1: 'On the importance and significance of Intermittency (Rebuttal of Section 3 of Rypdal and Rypdal 2016) by S. Lovejoy and C. Varotsos', Costas Varotsos, 03 Apr 2016

AC1: 'Reply to C. Varotsos-1', Kristoffer Rypdal, 07 Apr 2016

AC2: 'Results from the NorESM model', Kristoffer Rypdal, 09 Apr 2016

SC2: 'Trained eye deceived by fractal clustering', Costas Varotsos, 11 Apr 2016

SC3: 'Rebuttal of Section 2 of Rypdal and Rypdal 2016', Costas Varotsos, 13 Apr 2016

AC3: 'Reply to "Trained eye..."', Kristoffer Rypdal, 17 Apr 2016

AC4: 'Reply to "On testing the additivity..."', Kristoffer Rypdal, 17 Apr 2016

SC4: 'Final comment on low frequency linearity, nonlinearity', Costas Varotsos, 26 Apr 2016

SC5: 'Summary of Lovejoy and Varotsos rebuttal (SC1, SC2, and SC3) to Rypdal and Rypdal comment and replies (AC1, AC2, AC3, and AC4)', Costas Varotsos, 26 Apr 2016

In the end, the exchanges have veered far from the original issues, sometimes into sterile squabbles about what are the appropriate definitions of notions such as "Levy process" or "multifractal" - or the analysis by R+R of an entirely different numerical model (NorESM) - the results of which while being relevant to a wider scientific discussion - are irrelevant to the conclusions of the L+V paper that was in fact under debate. This was not helped by the fact that in the exchanges, L+V did not insist strongly enough on focusing the discussion on the basics, allowing the debate to become too far flung.

As a reviewer, I found these public exchanges a bit astonishing: recall that the entire debate is about the degree of linearity of outputs of two indisputably nonlinear numerical models (the Z-C model and the NASA GISS E2R model). The models are by construction nonlinear and L+V give prima facie evidence that the nonlinearity is indeed evident in the both the high frequency response to strong, intermittent signals such as volcanic eruptions, and the low frequency response to combined solar and volcanic forcings. From a scientific point of view, there is little debate about the fact that the climate gives a nonlinear smoothing of volcanic forcings, and that at long enough time scales that the responses to climate forcings are also strongly nonlinear. Therefore both L+V and R+R have posed the question backwards: the problem is not to find under which circumstances the model is nonlinear but on the contrary to show under what ranges of amplitudes and of time scales that the nonlinearity is weak - and to quantify its weakness. Unfortunately the debate spiralled away into technical issues that were of little relevance to this central question. Beyond that, I can see the frustration on both sides. Take for example the dispute about the effect of a linear response to an intermittent (multifractal) forcing – this basic result is now nearly thirty years old and is not hard to understand. If the forcing is from a multifractal process over a wide enough range of scales (i.e. the structure function exponent of the forcing $\xi$for(q) is indeed concave when determined by averaging over an infinite ensemble of realizations), that the the exponent of the response of a linear system (again, if linear over a sufficiently wide range of scales), can at best differ by a linear term i.e. the response

can at most be $\xi$resp(q)= $\xi$for(q)+qH where the H is the exponent of the linear system Green's function. Clearly, if the Green's function of the linear system was not scaling, then the response would break the scaling in which case the exponent itself is no longer meaningful. Hence, if the forcing and the responses of an infinite ensemble over an wide enough range of scales show a nonlinear difference $\xi$resp(q)$-\xi$for(q), then this can only be because the system is nonlinear.

R+R attack this result both mathematically and numerically. Mathematically, they impute a number of assumptions in particular the scaling of the transfer function. However, this assumption is not needed as an extra assumption since the statement "$\xi$resp(q)$-\xi$for(q) is nonlinear" is only meaningful if the forcing and response – and hence the transfer function – are indeed scaling (L+V only reject the need for extra mathematical assumptions). The (nontrivial) problem is therefore to gauge how confident we can be that the empirically/numerically estimated exponents $\xi$resp(q) and $\xi$for(q) are indeed representative of a process (i.e. of an infinite ensemble), and this over a wide enough range of time scales. This is the true weakness of L+V's claims. Their results are from over barely a factor of 100 in scale and from a single realization of the volcanic forcing – and are thus unsatisfactorily limited (as L+V more or less admit). Here, the numerical results of R+R concerning linear oscillators may be of some relevance. What R+R have shown is that for a single realization, over a range of $\approx 100$ in scale that one can concoct a linear process that comes surprisingly close to mimicking the response of the ZC model. But what does this prove? On the one hand – within the constraints of the available data - L+V give prima facie evidence that the system is strongly nonlinear, on the other hand R+R show that – due to these constraints – that L+V's conclusions might have been produced by an appropriately concocted linear system. The trouble is that one knows that - by construction - the numerical model in question is in fact nonlinear, and the mere fact that a linear model can be concocted that reproduces the results over a narrow range of scales and over a single realization in no way forces us to concludes that the model is instead linear!
My impression is that L+V conceded too little – e.g. they essentially ignored the linear oscillator result as being irrelevant – whereas R+R concluded too much – that this result somehow forces us to accept that the model that we know to be nonlinear is in fact linear. In this example and others, the difference between L+V and R+R often appears to be one of scientific methodology.

My conclusion is that the literature debate that started in the ESDD version of the L+V paper has ended up being more of a shouting match than a constructive exchange. Some interesting points have emerged - the most fundamental being that the model outputs that L+V analysed were only marginally adequate for their purpose – either to show the subaddivity at long scales or the nonlinearly volcanic response. But these points were more or less already conceded in the L+V ESD paper if they had been aware of a more appropriate, more convincing suite of model outputs, they surely would have used them. The extra comments in R+R mostly underlined the need to delve further - and I think that L+V would probably concur: the main point of their original paper was to pose the question "what are approximate conditions for linearity?" and to suggest methodologies for dealing with it.

Aside from this, a few interesting points did emerge – not so much from the from R+R paper itself - but from the ensuing exchange. I'm thinking in particular of R+R's new analysis of the NorESM model or the interesting finding by L+V that the Levy process model with independent spikes yields both realistic clustering (in spite of having no temporal dependencies), and that it is quantitatively close to both a cascade based model and to the real volcanic data. But these are best the subject of proper peer reviewed publications, not ESDD commentaries.

My overall recommendation is therefore that the new R+R paper does not add sufficiently to the already significant exchange on the original L+V paper. The question that they – and L+V - pose is posed backwards and the new comments in R+R add little to the questions already discussed in the ESDD debate on the original L+V paper. Instead, I encourage all the authors to spend their efforts clarifying, quantifying the

weakly nonlinear parts of the models and indeed of the real world!

---

## Author Comment (AC5) · 3 May 2016

This is a reply to reviewer #1

We agree with the reviewer that this discussion has digressed very far from the L&V paper and from our comment to it. But that is not our fault. Rather than sticking to the real issue, L&V started the discussion by publishing a very lengthy reply where the main message was that we were ignorant about the multifractal formalism. We had no other choice than responding to this attempt to discredit us as incompetent novices to the field. It is rather ironic that L&V in their last reply complain that we

are "too mathematical," whereas they are "physical." Our comment makes use only of elementary mathematics, we illustrate our points with simple demonstrations, and we mostly use the methods of and L&V and software downloaded from Lovejoy's web site.

The reviewer writes: "I found these public exchanges a bit astonishing: recall that the entire debate is about the degree of linearity of outputs of two indisputably nonlinear numerical models."

How can there be a degree of linearity? The meaningful question is: what is the degree of nonlinearity. And we never disputed that the models are nonlinear. The issue is if this nonlinearity is strong enough to be detectable in global temperature by the methods employed by L&V. The proper way to test this is to find data and methods to reject the linearity hypothesis.

The reviewer writes: "My impression is that L+V conceded too little – e.g. they essentially ignored the linear oscillator result as being irrelevant – whereas R+R concluded too much – that this result somehow forces us to accept that the model that we know to be nonlinear is in fact linear."

We find it very depressing that the reviewer, after all the exchanges on methodology in the discussion, can write that we conclude that the model is linear. This is wrong! We do not conclude that, neither in the comment, nor in the ensuing discussion. We have stressed again and again that linearity is a statement that cannot be verified (just like the statement that the photon has zero mass). This is not semantics, it is a fundamental principle in the philosophy of science. But linearity can be falsified by proper data and a proper test, and this is why the linearity hypothesis is a well-posed problem. Our whole point is that the two tests devised by L&V do not reject the linearity hypothesis. If nonlinearity cannot be detected by rejection of the linearity hypothesis in global temperature data, it gives credibility to linear modeling of the global temperature response. This is why it is important to clarify whether L&V's tests are correct or not.

In our Figure 2, we demonstrate that L&Vs test for subadditivity in the ZC-model is

invalid. As a reviewer, one of us pointed out the unsatisfactory way L&V dealt with this issue in their Figure 3 of their paper, and this was still unsatisfactory in their published paper. The demonstration in Figure 2 of our comment was not made in the review process, and this comment is our only possibility of publishing it. Neither the discussion, nor the reviewer, have disputed the correctness of our Figure 2. Hence, if ESD will not publish this demonstration the journal sends the false message that the peer-review has established that Figure 3 of the L&V paper is correct, and that our Figure 2 is wrong. Alternatively, it sends the message that the journal will not accept comments on papers published by influential scientists.

The problems connected with the invalid implicit assumptions (I-III) in the L&V paper were raised by us in the reviews, but rejected by L&V with highly unsatisfactory arguments. We should probably have recommended rejection, but there was obviously a need for a more in-depth discussion of these points. As an alternative, the editor suggested to write a peer reviewed comment, which we did. In Sect. 3.2 of our comment we demonstrate theoretically that imperfect scaling in response function and structure functions may give rise to different estimated intermittency in forcing and response, even if the response is linear. The reviewer does not point out any error in this section.

Nevertheless, we stress in our comment that Sect. 3.2 is not essential for our conclusion. The essential thing is the demonstration that a linear response model with internal noise can reproduce the trace-moment results of L&V. The reviewer buys the argument of L&V that this is demonstrated only by one realization of this linear model, and ignores that we have made the code available for anyone to check that this is a statistically robust result. If the editor invites us to submit a revision, we will include the results of an ensemble run which proves this point.

The most important effect that produces the observed change of intermittency between forcing and response is probably the high internal variability. It is quite astonishing that this effect is not commented by the reviewer. The main reason for including the data from the NorESM model was to demonstrate the crucial effect of that internal variability.

How can the reviewer ignore that!

That L&V neglect internal variability was raised in the review, but it was rejected by L&V as unimportant by very obscure arguments. We have demonstrated in our comment (and we will include the NorESM results in a revised version) that it is very important. If the reviewer cannot prove otherwise, and still recommends that it is not worth publishing as a comment, the reviewer in fact recommends ESD to refrain publishing corrections to incorrect published results.

The reviewer ends his report by encouraging all the authors "to spend their efforts clarifying, quantifying the weakly nonlinear parts of the models and indeed of the real world!" That is an issue we are already working on with data from the NorESM model. However, those nonlinearities are difficult, and maybe impossible, to detect in the global temperature series. The likelihood of detection is much greater in regional or local temperature data, and in other climate variables. One cannot quantify nonlinearity if one is not able to detect it. And detection means rejection of the linearity hypothesis.

The reviewer seems to be of the opinion that it is unimportant whether the tests devised by L&V to detect nonlinearity are valid or not – since the models obviously are nonlinear anyway. Then one cannot avoid asking the crucial question; what was the point of publishing the L&V paper in the first place?

---

## Short Comment (SC6) · 5 May 2016

The comment was uploaded in the form of a supplement:
http://www.earth-syst-dynam-discuss.net/esd-2016-10/esd-2016-10-SC6-supplement.pdf

---

## Author Comment (AC6) · 5 May 2016

This comment by Varotsos and Sarlis suffers from severve logical weaknesses.

Reply to point 1: We reproduce the result that Varotsos and Sarlis (V&S) refer to in Fig. 1 of our response named AC4, and demonstrate by a Monte Carlo simulation that the difference between the two curves are statistically insignificant. We did not discuss this in our original comment, because we used the entire 1000 yr record to construct a signal for the internal variability, and got very small difference between the curves. LSV 's use of the first 195 yr of volcanic forcing response as a measure of internal variability

is of course legitimate, but it comes with a penalty of greater statistical uncertainty.

Reply to point 2: V&S misses the point that the finite-sample size uncertainty for a 1000 yr time series is much smaller than the uncertainty for the 195 yr series. But V&S' main problem is that they continue to be utterly confused about the logic of the tests (and so does Reviewer #1). All our tests are devised to REJECT the linearity hypothesis. It is quite obvious that the curves corresponding to the red and brown bullets in Fig. 1d of our comment are so close that they are within the confidence limits that would result from the same kind of Monte Carlo simulations that we did in AC4. This means that the test does not reject linearity.

Reply to point 3: We agree that the main effect that leads to the difference of inter-mittency between forcing and response in our harmonic oscillator example probably is caused by the internal variability. But so what? If this is the case, the test demonstrates that internal variability explains the trace-moment results of L&V in a linear response model. This is an important and highly relevant observation.

V&S argue that they have falsified the linearity hypothesis in points 2 and 3 above, and therefore our demonstration concerning the trace moments is irrelevant. As discussed above, we don't accept that the linearity hypothesis was falsified in points 2 and 3, but let us for the sake of argument assume that it were true, and let us look at the original L&V-paper. They first devised a test of subabdditivity which they contended rejected linearity. Then they devised the trace moment analysis as an alternative test. According to the logic of V&S, the trace moment test of L&V should then be irrelevant. So why publish it?

---

## Author Comment (AC7) · 7 May 2016

R&R's summary of the discussion

The discussion has followed two almost separate paths:

Path1: SC1->AC1->SC2->AC3->SC5->RC1->AC5

Path2: AC2->SC3->AC4->SC4->RC1->AC5->SC6->AC6

Path 1 was initiated by L&V who wanted to continue a discussion about "intermittency in general and volcanic intermittency in particular," which was started in the review

process of our discussion paper (now published in ESD) "Late Quaternary temperature variability described as abrupt transitions on a 1/f noise background," http://www.earth-syst-dynam.net/7/281/2016/

The discussion along this path is only weakly connected to the issue of nonlinearity in the global temperature response, and is quite technical. Of relevance to the L&V paper and our comment to it are two conclusions from our side:

(i) Internal variability can explain the entire difference between the intermittency estimated from the forcing signal and the ensuing temperature signal. This was demonstrated through our harmonic oscillator example in the comment, and through the structure function analysis of the NorESM data demonstrated in Figure 1 in AC3.

(ii) Lack of scaling in the response function, and in the structure functions, will lead to "spurious multifractality" and give rise to different intermittency estimates for forcing and response, even if the response is linear. The intermittency estimates will depend on the scale interval used for fitting a straight line to structure functions/trace moments in log-log plots.

Both (i) and (ii) can potentially explain the estimated difference in intermittency observed by L&V, but the NorESM analysis shows that (i) is sufficient. It is also a phenomenon that is intuitively very easy to understand; even if the intermittency in the forcing and response is the same, addition of a strong internal noise to the response will reduce its intermittency. Neither L&V, nor Reviewer #1, have presented any compelling evidence for the assertion that this difference is caused by nonlinearity in the response. Their claim that the demonstration made by means of the harmonic oscillator response is the result of a cherry-picked realization is easily rebutted. In a revision of our comment we will show this through a Monte Carlo study, and we will also demonstrate the effect by using a linear power-law Greens function rather than the harmonic-oscillator Greens function.

Path 2 started with our demonstration that our Figure 2 in the comment, which dealt

with the alleged subadditivity in the ZC-model, could be reproduced also in the NorESM model. And it also showed the high power on all scales in the internal variability as compared to the forced variability. L&V never contested these results, but in SC3, Varotsos, Sarlis, and Lovejoy (VSL) presented a "rebuttal" of another test we made in Section 2.3 of our comment article. Here they used the first 195 yr of the volcanic forced temperature time series to estimate the Haar fluctuation curve of the internal noise. The linearity hypothesis would have been rejected if this curve could be shown to be different from another estimated curve. The crucial issue here is whether or not the difference between these two estimated curves is larger than the statistical uncertainty of the estimates. We demonstrated in Figure 1 in AC4 that this uncertainty is considerably larger than this difference, and hence that the result is statistically insignificant and does not reject the linearity hypothesis.

In SC4, and reiterated in Point 1 in SC6, Varotsos, Lovejoy and Sarlis claim that the two curves are different and that we have not addressed this result. They simply ignore that we have demonstrated that the result is statistically insignificant. The discussion in SC6 demonstrates again that L&V are utterly confused about how to deal with statistical uncertainty in hypothesis testing. They don't understand that in all these tests of subadditivity, the test is to check if we can reject the statement that two Haar fluctuation curves are equal. This can be done if the difference between the curves is larger than the finite sample size uncertainty of the estimated curves. In all the tests under discussion this difference is smaller than the uncertainty, and the rejection of linearity fails.

In SC4 they manage to write that our presentation of results from the NorESM model is "an indirect admission that our original paper was correct." This absurd statement is another example which demonstrates that L&V dismiss basic principles of scientific methodology. So let us reiterate: a well-posed scientific hypothesis cannot be verified, only falsified. Linearity in the response is such a well-posed hypothesis. That the tests devised by L&V have failed to falsify linearity, does not mean that one should not stop

searching for new data and new tests that could falsify it. The larger the arsenal of data and tests which fail to falsify linearity, the greater our confidence in the hypothesis. It is therefore highly appropriate and relevant in a comment to bring in other relevant data and test methods.

The comments of Reviewer #1 (RC1) did not address any of the issues discussed in Path 2. The reviewer recognizes that there seems to be a disagreement on scientific methodology between us and L&V, but expresses the view that such a disagreement somehow does not belong in a public discussion and does not deserve to appear in a peer reviewed comment. We disagree with that view, and hope it is not the editorial policy of ESD. Discussion about scientific method and how papers should be written in a concrete context should be a central part of the open review process adopted by Copernicus journals.

---

## Short Comment (SC7) · 8 May 2016

**Summary of the discussion**

**S. Lovejoy[1], C. Varotsos[2], and N. Sarlis[2]**

[1]Physics, McGill University, 3600 University St., Montreal, Quebec, Canada

[2]University of Athens, Dept. of Physics, University Campus Bldg. Phys. V, 15784 Athens, Greece

Let us summarize our view of the discussion:

Someone (Lovejoy and Varotsos, 2016) proposes a scientific hypothesis and tests it against the data. They claim agreement between their hypothesis and the data with some level of *scientific* confidence, recognizing a) that *scientific confidence* is usually not quantifiable, b) that rigorous statistical hypothesis testing requires numerous additional hypotheses and assumptions, c) that such statistical testing can at best reject false hypotheses, never accept true ones.

Rypdal and Rypdal (2016) don't like L+V's scientific conclusions, so they concoct an alternative hypothesis (often pulled out of a hat without any attempted physical or scientific motivation, but no matter). They then perform a statistical test on their new hypothesis showing that it cannot be statistically rejected at some supposedly high level of statistical confidence. They then conclude that - since their alternative hypothesis cannot be rejected – that it must therefore be triumphantly accepted. Since the original L+V scientific hypothesis is logically incompatible with their new hypothesis, accepting the new one implies a rejection of the original one.

This is a complete abuse of both the scientific method, and also statistical hypothesis testing. With this approach– as we indicated in one of our previous responses - one can prove anything one wishes, including that the speed of light is infinite!

**References:**

Lovejoy, S., and C. Varotsos (2016), Scaling regimes and linear/nonlinear responses of last millennium climate to volcanic and solar forcings, Earth Syst. Dynam., 7, 1–18, doi:10.5194/esd-7-133-2016.

Rypdal, K., and M. Rypdal (2016), Comment on "Scaling regimes and linear/nonlinear responses of last millennium climate to volcanic and solar forcings" by S. Lovejoy and C. Varotsos, *Earth Syst. Dynam. Discuss.,* doi:10.5194/esd-2016-10, 2016.

---

## Author Comment (AC8) · 10 May 2016

We use our privilege as authors of the paper to have the last word after the discussion is closed. We would have preferred to be silent at this stage, but the summary of LVS contains so many incorrect representations of our views that we have to comment on them.

LVS summary: "L&V proposes a scientific hypothesis and tests it against the data."

R&R reply: What is this hypothesis? Is it that the response is nonlinear? In that case we have to reiterate our assertion that this is not a falsifiable hypothesis, because a

nonlinearity can be so small that it goes under any radar. The testable and falsifiable hypothesis is that the response is linear. If the linearity hypothesis is falsified, then the response is nonlinear. So, nonlinearity is verifiable, but not falsifiable.

LVS summary: L&V claim agreement between their hypothesis and the data with some level of SCIENTIFIC confidence.

R&R reply: We have acknowledged the concept of scientific confidence. It is built via theory that results in a falsifiable hypothesis, and a succession of increasingly sharp tests which are designed to falsify the hypothesis. In the context of our discussion, the only possible falsifiable hypothesis is that the response is linear.

LVS summary: R&R don't like L&V's scientific conclusions so they concoct an alternative hypothesis (often pulled out of the hat without scientific motivation..).

R&R reply: We have written nothing about our emotions concerning L&V's scientific conclusions. This kind of arguing has no place in a scientific discussion. There is no new alternative hypothesis. We just formulate the L&V hypothesis in a way that is falsifiable. We have presented our scientific motivation in Section 1 in AC4, it has been presented in several of our recent published papers, and it is the main subject of an upcoming paper of ours on linear "multi-box" energy balance models. The physical paradigm is that the global temperature response is governed by energy exchange between subsystems with different response times, and that this multitude of response times gives rise to the apparent scaling in the response. Energy exchange between subsystems can be a linear function of temperature differences, which results in a linear model for the global temperature response, and such a linear model allows predictions of linearity in the response that are falsifiable.

LVS summary: They then conclude that – since their alternative hypothesis cannot be rejected – that it must therefore be triumphantly accepted.

R&R reply: L&V's (and Reviewer #1's) repeated caricature of our views does not make

it more true. We have stated again, and again, and again in the original comment and in the discussion that failure of rejection of the linearity hypothesis by a given test and a given data set does make it "triumphantly accepted." The linearity hypothesis cannot be verified. Again, this kind of argumentative approach to the discussion by LVS is far below the standards that we expect in a scientific debate.

LVS summary: Since the original L&V scientific hypothesis is logically incompatible with their new hypothesis, accepting the new one implies rejection of the original one.

R&R reply: Assuming that "the original L&V scientific hypothesis" is that the response is nonlinear, and the "new one" is that the response linear (the "original" is the negation of the "new"), it is of course true that accepting the new implies rejection of the original. What LVS fail to understand is that one cannot construct a test to accept a hypothesis, because a well-posed hypothesis cannot be verified, only falsified. We have not devised a test to "accept" linearity, but a test to reject it. The logically correct way of formulating LVS' statement would be: Rejection of the linearity hypothesis implies acceptance of the nonlinear hypothesis. And from the law of induction: Repeated failure of rejecting linearity by increasingly sharper tests leads to greater confidence in the linearity hypothesis.

LVS: This is a complete abuse of the scientific method, and also of statistical hypothesis testing. With this approach – as we indicated in one of our previous responses, one can prove anything one wishes, including that the speed of light is infinite.

R&R reply: This is a very strong, and absurd, statement. The hypothesis that the speed of light is infinite is a well-posed and falsifiable hypothesis, which has been falsified in experiments. By falsifying this hypothesis we have verified that the speed of light is finite. The hypothesis of zero photon mass, which was another analogy mentioned by L&V in an earlier comment, is also a well-posed hypothesis, which has not been falsified by experiments yet. The hypothesis-testing method that we devise works perfectly well for both of these examples. The zero photon mass is a good

analogy to the linearity hypothesis. It is possible that the photon mass is finite, but that it is so small that it has so far gone under the radar. The continuing failure to detect a finite photon mass by experiments that seek to reject the zero-mass hypothesis, gives us great deal of confidence in this hypothesis.

---

## Referee Comment (RC2) · Anonymous Referee #2 · 22 May 2016

The Lovejoy & Varotsos (L&V) paper that this is a comment on has some issues that this comment on it identifies; I thus think it is essential that this comment be published, although it does need some minor revisions.

1. I started with re-reading L&V, and some brief comments are in order. First note that, in my reading of L&V, there are many places where statements are made that are flatly not true or highly misleading, as well as the presentation being unnecessarily confusing and imprecise. With a particular beef about their failure to carefully distinguish between (i) whether the underlying system equations are nonlinear, which

no-one would disagree with, and (ii) whether the response to small-amplitude external forcing can be sufficiently well approximated by the linear response for many purposes. No-one would ever claim that linearity is "valid", but simply that it is a sufficiently good approximation for some purposes. (And as an aside, regarding the observation on page 143 that a linear scaling system corresponds to a filter that is a power law... since we know that the filter is not a power law, then for this sentence to be true the scaling assumption must not be valid, invalidating the analysis. And I also object to sloppy usage of the word "feedback" which they tend to incorrectly invoke as an explanation for nonlinearity.) Personally, I would never have recommended acceptance of L&V manuscript without major modifications.

2. In addition to the observations about the L&V presentation being unnecessarily confusing and that the conclusions there are not actually supported by the analysis, I do think it worth pointing out somewhere that the ZC model is wholly inappropriate to the addressing the question in the first place. This 1986 model was designed solely to capture ENSO dynamics, and 30 years of subsequent research has made it clear that the parameter values assumed in the 1980's were not correct. The model is not stable, and a self-sustained ENSO arises as a result of chaotic dynamics (see papers by Tziperman in 1990's), rather than being the result of a stable heavily damped oscillator driven by climate variability as most researchers now believe ENSO dynamics result from in the real climate system. The characteristics of variability in the ZC model are therefore not relevant to reality (nor the behavior of GCMs) as it wasn't designed to capture them in the first place – I would not have been surprised if the ZC model is nonlinear in its response to forcing, but I don't actually care one way or the other. It is a complex but toy model, not a GCM intended to capture the dynamics of the actual climate. (Also, as a result, variability in ZC may have nothing to do with variability in GISS. I could generate a long control run for you if you really wanted to assess unforced variability.)

3. Some of the criticisms of L&V seem too directed at the individuals rather than describing the paper itself (e.g. "L&V are blind to this fact"). I understand your frustration with their paper, but criticize the paper, not the authors.

4. Citations on line 28-29, could add MacMynowski et al (2011), where the transfer function to solar forcing was explicitly computed in a GCM, Myrrhvold & Caldeira (2014), where they fit the response of CMIP5 models to either semi-infinite diffusion (long-memory power law) or multi-exponential, or Held et al (2010) with a two characteristic response time in GFDL models.

5. Could also be a bit clearer in places about "linearity". We all know that the response isn't exactly linear, the question is whether it is sufficiently linear for our purposes. E.g., line 35, these don't demonstrate that CMIP5 is linear, simply that it is approximately linear for the amplitude of forcings considered, and for estimating the forced response (which is a different question from asking whether the forcing alters the statistics of the variability, as the authors note above eq. 1). Again, line 149 ought to have a qualifier like "substantively influence" or something like that; of course it will to some level that we are hoping we can ignore for most purposes. It is clear from the authors' response to the online discussion that they understand what they mean, some of the clarity in their responses may be useful in the paper.

6. Top of page 3, point (i), need to say "the distribution of the internal variability" or something like that (the actual realization will of course change.)

7. Page 4, 5, some additional discussion is missing; in order to separate out the variability from the forced response, you either need to make some additional assumptions (as you do) that the high-frequency content is only variability and not forced (i.e., that the transfer function from input to output is small at high frequencies, which is reasonable), or actually estimate the input-output response, which would be difficult due to limited data. The linear response to solar forcing could be frequency dependent, rather than the functional form in equation (8). In principle, one could subdivide the time series and estimate (nonparametric) transfer function fit to the response (e.g. MacMynowski

& Tziperman, GRL, 2010), or do a least squares fit of the time series to a more complicated (but a priori) linear model with more than the two parameters you have (even AR(1) I would believe more than what you use in (8).) And use that to estimate the residual variability. This is highly unlikely to alter any conclusions, so I don't object to what you did, other than to more explicitly mention in your discussion that the separation into forced response and variability requires an assumption on the nature of the linear input-output relationship. (And note that you already described some of the literature on that relationship only a page earlier!)

8. Figure 1d and 2, is it easy to put confidence bounds on these? (Without the plot becoming too excessively messy.) (I know you said you didn't need to bother, though if it were trivial to do it might add weight.)

9. Sentence on page 6, line 174, needs to be fixed... I would generally assume "the statistics are so poor" to mean that the differences are NOT statistically significant.

10. Line 201 might be a good point to point out that the ZC model was never intended to get the statistics of variability correct, and so there is no basis for assuming anything about the magnitude of it relative to GISS. (If I can get it to work, still, I could go generate some ZC runs for you... I do have a 1000-year output file still that is probably from the right parameter values, though not absolutely certain; I'll try posting that as a supplement).

11. Line 224, not clear that it contradicts L&V insofar as one could get the wrong variability at low frequencies for different reasons – nonetheless it clearly contradicts everything we know about the climate response to dynamic forcing, you already have lots of citations earlier on this point (and the MacMynowski 2011 GRL paper quite explicitly computes the transfer function for one GCM and shows that it is not a power law). So I think you can be pretty strong here in pointing out that this assumption in L&V is simply wrong. (I find it astonishing that L&V would make such an obvious error, yet it appears from my reading of the manuscript that they do.) Point III is also strong.

I don't feel qualified to argue regarding point #2

12. Typo line 264 (of vs on)   I will also make some brief comments on the back-and-forth "discussion" between R&R and L&V, typed chronologically as I skim:

13. (SC1,AC1). L&V criticize only the second of the 3 issues that R&R raise regarding L&V's analysis; they appear here to ignore the other two. They then go on to claim R&R make an error when, if I read correctly, R&R simply have reverted to the usual structure function definition rather than the abnormal one L&V use. I don't see anything in here that would suggest that changes to the manuscript are needed, though some of the authors' comments in response may be helpful in clarifying the manuscript. (I think the phrase, "The memory in the response smears out the volcanic spikes. This is a linear effect." might be useful to include somewhere, along with some of the opening comments.)

14. (SC2). a. If I didn't know that it was the same people, I would have thought from the initial sentence here that its authors' hadn't even read L&V let alone R&R, insofar as the original paper reads almost entirely as an argument that approximating the system response as linear is not a good approximation, while the first sentence here would have one believe that the original paper was simply pointing out the (obvious) idea that it isn't precisely linear. R&R simply observe that one can't statistically rule out the response being linear (i.e., can't distinguish it from the response of a linear system), which seems to me like a pretty good argument that linearity is a sufficiently good approximation, they never claim that the response actually is linear. b. The statement that L&V's first response indicated assumptions I-III as being irrelevant is utterly false, the first response only stated that assumption II was irrelevant and ignored the fact that there were two others, which are both relevant. Further, #I was clearly stated as an assumption (albeit one not actually satisfied by the climate system as I noted in point 1 above) in the original L&V paper. Most of this note seems to not actually respond directly to the comment under consideration, and certainly doesn't seem to refute any results in that comment. c. Minor comment #1 seems to utterly misconstrue the intent

of R&R's comment. It would seem that some clarifications in the comment might be in order in case some other readers are also unable to detect the distinction between "the response is linear" and "the response is not statistically different from that of a linear system and therefore the linear approximation is sufficient for most purposes". I thought the wording in R&R could have been improved in places, but nonetheless thought this point was blindingly obvious.

15. (SC3) a. Minor criticism, but the presence of temperature-albedo feedbacks does not in and of itself suggest that the response to forcing will be strongly nonlinear. b. And again, this comment fails to distinguish between the concept of the response being nonlinear, which is trivially obvious and no-one disagrees with, and the response being significantly different from linear, i.e., whether or not using a linear approximation is sufficient. c. I agree that the NorESM analysis does not directly argue whether L&V's original analysis was correct or not.

16. SC3, SC4: a. I'm not going to dig in myself as to why the analysis in R&R regarding ZC noise levels appears to be inconsistent with the SC3 computation of noise levels, either one of them is wrong, or there are different definitions going on here. (I see R&R subsequently responded to this.) b. I will comment on the statement: "We take their AC3 response (R+R, 2016c), - the analysis of a completely different set of model runs (NorESM) – to be an indirect admission that our original paper was correct." While I am not formally reviewing the discussions, I find this passive-aggressive behavior to be childish and inappropriate.

17. SC5 – this adds nothing other than reinforces my conclusion that it's authors did not actually read or understand R&R.

Bottom line: Having read most of the discussion, I find (a) almost all of it is utterly irrelevant to the R&R comment under consideration (as both parties agree), and more importantly (b) none of it appears to refute the conclusion in R&R that a properly constructed test without arbitrary and erroneous assumptions does cannot distinguish

the response from linearity. I therefore see nothing in the posted discussion that lead me to worry that I missed something in my reading of the R&R response.

Please also note the supplement to this comment:
http://www.earth-syst-dynam-discuss.net/esd-2016-10/esd-2016-10-RC2-supplement.zip
* * *

---

## Author Comment (AC9) · 23 May 2016

Thanks for a constructive review, containing many useful suggestions for improvement of the paper. If the editor approves submission of a revised paper we will take the reviewer comments into consideration. We may also use the data from the Zebiak-Cane model control run given in the Supplementary file of the reviewer, but have one question. The data used by L&V are averaged over 100 realisations. Is the data given in the file the result of one single control run, or is it averaged over an ensemble of realisations?

---

## Referee Comment (RC3) · Anonymous Referee #3 · 27 May 2016

The Rypdal and Rypdal manuscript (R&R) raises several critical points about the Lovejoy and Vartsos (L & V) paper, and my recommendation is that it should be published with minor revisions.

General comments:

The original L&V paper broaches important points about the scaling of forcing and response over long time scales. However, it suffers from major flaws both of substance and style, the latter being beyond the scope of this review.  The R&R comment addressses some of the substantive flaws, with their critique seemingly centered on three issues:

(1) A hypothesis test fails to reject the null hypothesis of linearity.
(2) Properly accounting for natural variability reduces the perceived degree of non-linearity
(3) Distinct intermitency of forcing and response may still be consistent with a linear system.

In my assessment, points (2) and (3) are well founded, with some specific comments detailed bellow. Point (1) however merits a more careful discussion.

[1] The response functions of the CZ and GISS models, as well as the climate system, are non-linear in the strictest mathematical sense (e.g. the Black Body/Plank Feedback is non-linear). Thus, the appropriate question to ask is whether a linear approximation is valid for a certain purpose. L&V seem to ask the question as to whether a linear approximation is valid when assesing the mean temperature response to small perturbations in radiative forcing, of the type expected to arise from historical changes in volcanic aerosol emissions and solar variability.  This is an important question for purposes of inferring climate sensitivity from palaeorecords, and L&V seem to reach the conclusion that the linear approximation is not appropriate, as the response is markedly sub-additive, by a factor of R~1.5. L&V do not set up the problem as such a null-hypothesis, and I agree with R&R that this is a major flaw.  They also fail to quantify the uncertainty range of their factor R. Although unlikely, it might be that 1.5 is consistent with a linear approximation in the presence of noise.

However, issues of presentation and robustness aside, the L&V paper can be easily interpreted as a rejection of the null hypothesis of linearity. While R&R perform a more properly set-up hypothesis test, it is a somewhat different hypothesis test, examining the second order statistics of the residuals of a specific model fit, as opposed to the responses themselves.  Thus, between L&V and R&R, a null hypothesis has been proposed, and two different tests have

been performed, one of which rejects it, while the other one fails to reject it. This issue needs to be adressed before the R&R test can be interpreted as a rebuttal of the L&V result. If the R&R test implies that the L&V test should have also rejected linearity, then the question arises where is the error in L&V? (perhaps not including internal variability?). Otherwise, the alternative test is not by itself a rebuttal.

[2] At times the language used is not appropriate for a scientific paper. I recommend that the critique sticks to the sciences and does not become personal and subjective.

- Line 164: "invalid (and completely unnecessary) approximation". 'Completely unnecessary' sems a bit hyperbolic.
- Line 164-165: "The authors admit in the published paper that this analysis is wrong". I think R&R misrepresent L&V when claiming that the latter explicitly admit to a faulty analysis.
- Line 318: "but L&V are blind to this fact". This phrasing is inappropriate and, I dare say, completely unnecessary.

More Specific Comments:

[2] Line 31: Geoffroy et al (2013) finds that the linear approximation is appropriate when considering two different forcing scenarios with the same type of forcing ($CO_2$). Even then, in many of the models a small but robust overestimation of the temperature response in the 1pctCO2 scenario can be observed. There is also evidence (Merlis et al 2014) that there is a small but noticeable difference in the response to volcanic forcing as opposed to CO2 forcing.

[3] Line 37: The 'nonlinearity' that Andrews et al (2012) find refers to the fact that the relation between global temperature and global radiation imbalance is "not a line" in the transient regime. This may be entirely consistent with linear dynamics (Armour et al 2013).

[4] Line 115 (equation 8). An AR(1) type transfer function would be easy to fit and more consistent with a dynamical system than using a constant lag. Additionally, as per Geoffroy et al (2013), using at least two time-scales to represent a fast and slow component seems to be a minimal requirement for a decent fit. Such a fit should be easy enough to perform given knowledge of the forcing (e.g. Castruccio et al 2014). However, for the author's purposes equation (8) should be sufficient, and I would not list this as a required improvement.

[5] Along the same lines: How different are the transfer functions fit to the solar only, volcanic only, and solar+volcanic scenarios? This could be construed as yet another hypothesis test.

[6] Lines 153-155: A visual comparison seems unsatisfactory as a hypothesis test. A more rigorous p-value test would be nice. However, as L&V do not provide one either, I do not think that the R&R comment should be held to a different standard.

[7] Section 2.4: Lines 157-180. Confusing. What is the invalid and completely unnecessary approximation? An alternative estimate is given, but no explanation for why the original estimate was wrong when variability is not taken into account.

[8] Lines 224-226: In support of this, the authors can cite the Geoffroy et al (2013) also cited in the introduction, which provides estimates of the transfer functions for CMIP5 gcms (appendix of part I). Additionally, MacMynowski et al. (2011) explicitly compute the transfer function for solar forcing, showing that it is not a simple power law.

[9]Line 331: A damped harmonic oscillator is an unusual choice for a transfer function. The response to volcanic forcing is generally assumed to follow a standard decaying exponential form. One could use a standard transfer function such as those mentioned in [8].

Armour, K. C., Bitz, C. M. & Roe, G. H. (2013). Time-Varying Climate Sensitivity from Regional Feedbacks. J. Climate 26, 4518–4534 (2013).

Castruccio, S., McInerney, D. J., Stein, M. L., Liu Crouch, F., Jacob, R. L., & Moyer, E. J. (2014). Statistical Emulation of Climate Model Projections Based on Precomputed GCM runs. *Journal of Climate*.

Merlis, T.M., Held, I. M., Stenchikov, I. M., Zeng,F. and. Horowitz, L. W. (2014). Constraining Transient Climate Sensitivity Using Coupled Climate Model Simulations of Volcanic Eruptions. *Journal of Climate.*

MacMynowski, D. G., Shin, H.-J., and Caldeira, K.., The Frequency Response of Temperature and Precipitation in a Climate Model*. Geophysical Research Letters.*

---

## Author Comment (AC10) · 21 Jun 2016

We thank the reviewer for illuminating comments, which had been incorporated in the revised paper. Below follows a detailed response to the comments.

[1] On the hypothesis testing for subadditivity. We perform two tests for the subadditivity of the ZC model. One, described in Sect. 2.2 and 2.3, is a different test from the one performed by L&V, since it includes internal variability (which L&V do not). The reviewer is quite correct in pointing out that this test does not represent a direct rebuttal of the L&V test, if one assumes that internal variability can be ignored. However, in Sect. 2.4

we repeat the L&V test (i.e., without including internal variability) and present the result in our Fig. 2. We don't find the factor 1.5 claimed by L&V on the longest time scales, so this figure presents a rebuttal of the L&V test. See also our response to Point [7].

On the language - personal and subjective. Lines 164-165. We think it is important to point out that the approximation in question is unnecessary. This is neither personal nor subjective. Approximations are justified if they simplify things, and do not introduce biases. In this case, the approximation does not simplify anything (computations are just as easy without it), and it introduces a bias towards subadditivity. In their revised paper L&V present results both with, and without, this approximation (see their Fig. 3), which demonstrates this bias. We take that as an admission that the analysis based on this approximation is wrong (biased). However, they don't draw the obvious conclusion and omit the approximate analysis and results, but present both as two alternative approaches, and in the concluding section they present the difference between the approximate and exact result in a way that misleads the reader to interpret it as an uncertainty range. We think it is appropriate to point out these facts, but in the revision we have reduced this paragraph to pointing out the nature of their approximation, a brief description of the results they have presented, and a description of our findings.

Line 318: In the revision we have removed the the offending phrase, which we agree is unnecessary.

[2] Line 31: In 8 out of 16 models studied by Geoffroy et al. (2013) one can observe a very small overestimation of the transient response in the 1 pctCO2 scenario when parameters in the two-box model are estimated from the 4xCO2 step-function scenario. This discrepancy does not have to arise from nonlinearity, however. It is just as likely a result of the simplicity of the two-box model. It is well known that a long-memory response will lead to a slower temperature rise under transient forcing than a short-memory response (Rypdal and Rypdal, 2014, Rypdal, 2016). The physical reason is that a long-memory response is associated with energy transport from the surface into the abyss and hence slower temperature rise at the surface. Hence, if the GCMs

contain a response on even longer time scales than the long scale in the two-box model the result would be a slower temperature rise in the GCMs than in the two-box model for the 1pctCO2 forcing.

As we understand Merlis et al. (2014), volcanic forcing and abrupt CO2 change yield similar values for the fast component of the climate sensitivity in GCMs, but 5-15% smaller than the transient climate sensitivity. For the same reason as explained above, long memory in the response will give rise to a lower transient response and an under-estimation of the sensitivity. Hence, these effects do not necessarily imply nonlinearity in the response.

[3] Agree, see e.g., Fig. 8 in Rypdal and Rypdal (2014). We have decided to omit the reference to Andrews et al. and include several others that are more relevant.

[4] For global GCMs we know that a two-exponential, or a power-law, response function work quite well, and we have a pretty good ide why. It has to do with the different thermal inertias of the mixed layer and the deep ocean, and the rate of heat exchange between the two. We have much less clear ideas about the response function of the ZC-model. As Reviewer #2 pointed out. The ZC model is very different from a GCM. The 25 yr time delay response to the slow solar forcing is solely based on visual in-spection of the forcing and response time series is admittedly very crude, but we have no reason to believe that a more sophisticated response function is any better. Since the purpose here is just to find an estimate of the variance of the internal variability, we think the approach makes sense.

[5] Along the same lines. The reason why we cannot do this in a meaningful way is that we have so poor knowledge about the response function for the ZC model on the short time scales. The reason we chose a harmonic oscillator model in Fig. 4 is the apparent enhanced ENSO oscillations after major volcanic eruptions. If we use a certain response function and we get different fits for the sum of responses from the responses to the sum of forcing we can always blame the incorrect response function.

Hence, this will not construe another test.

[6] In principle we agree that we should put confidence intervals on these two curves to demonstrate that they are not significantly different from each other. This could easily be done as we do in Fig. 2 of the revised paper by Monte Carlo simulation of 1/f processes. However, in Fig. 1d the two curves to compare are so much on top of each other that they cross each other several times. We have explained this in the revision.

[7] This point was discussed in our response to point [1], but let us elaborate on it here. The "invalid and completely unnecessary approximation" would be apparent by reading Sect. 3.4 in the L&V paper. The approximation is the basis of their Eq. (5), which assumes that one can neglect a cross term which is the product of the solar response and the volcanic response on a given time scale $\Delta t$. L&V argue that one can do this because solar and volcanic forcing are statistically independent processes. The approximation would have been OK if we had a large ensemble of realisations of solar and volcanic forcing to average over, but in this case we have only one realization of each (the historic forcing over the last millennium). One of us (K. Rypdal) was a reviewer of the paper and pointed out this weakness in the first review. The result was that L&V kept the old results, but added a paragraph at the end of page 8 where they admit that "the cancellation of the cross terms assumed by statistical independence is only approximately valid on single realizations, especially at low frequencies where the statistics are worse."

The source of this error is probably rooted in the sloppy notation of using angular brackets <> for averages which are really not ensemble averages (or expectations) but rather estimates in the form of time averages. If two quantities X and Y are statistically independent their the expectation E[XY]=0, but the time-average estimate <XY> is normally nonzero, and on long time scales $\Delta t$ we have have virtually no statistics, so there is no reason to believe that <XY> is a good estimate of E[XY]=0.

To us it seems clear that L&V have understood the error, and the appropriate response

would be to omit this approximation and replace the blue curve in their Fig. 3b with the one computed without this approximation. But then this Fig. 3b would look similar to our Fig. 2, and obviously be much less convincing. Instead they present the "correct" curve as a ratio given by the lower curve in their Fig. 3c, along with the "incorrect" ratio (the upper curve). The "correct" ratio is probably more or less the same as we would get if we compute the ratio between the red and the blue curve in our Fig. 2 (our results are not completely identical to L&V, which may be due to slightly different steps between the values of $\Delta t$ where the Haar fluctuation is computed – but we have used codes downloaded from Shaun Lovejoy's web site). We also find that the red curve is higher than the blue for 200< $\Delta t$ <1000 yr, but on these time scales the fluctuation level is estimated from 5 effective data points for $\Delta t$=200 yr, and for only 1 effective data point for $\Delta t$=1000 yr. Actually the number of effective data points is even smaller because the time series is not white noise, but exhibits dependence on all scales. Hence, it is obvious that these differences are not statistically significant.

So there are two issues here. One is scientific; the results without the approximation are not significant. The other is the way L&V are presenting their results. In the revision we have decided not to dwell too much on L&V's presentation and focus on the results.

[8] We included the reference to MacMynowski et al., Geoffroy et al., and Fredriksen et al., which have presented spectra for a large number of CMIP5 models.

[9] One should keep in mind here that the harmonic oscillator response was employed for comparison with the ZC model, which responds very differently from GCMs. As stated in the paper, the purpose of this demonstration was not to present a realistic response model for either of the model results analysed by L&V. It was simply to demonstrate that the effects which L&V attributes to nonlinearity is easily produced in linear response models with internal noise. And as a pedagogical tool, we think a driven, damped harmonic oscillator is an excellent choice.

---

## Author Response (AR2)

**Changes in second revision**

The discussion section (Sect. 4) has been removed. Some material related to the physical basis of nonlinearity in the response has been moved to the final section (Discussion and conclusions), and is marked in red.  All of this material refers to the L&V paper and not to the interactive dicussion, and cannot be considered to be "tangential to the main arguments."